# Master–Slave Game Optimal Scheduling for Multi-Agent Integrated Energy System Based on Uncertainty and Demand Response

**Boyu Zhu** * and **Dazhi Wang**

School of Information Science and Engineering, Northeastern University, Shenyang 110004, China; wangdazhi@ise.neu.edu.cn
* Correspondence: z1838577855@163.com

**Abstract:** With the transformation of the energy market from the traditional vertical integrated structure to the interactive competitive structure, the traditional centralized optimization method makes it difficult to reveal the interactive behavior of multi-agent integrated energy systems (MAIES). In this paper, a master–slave game optimal scheduling strategy of MAIES is proposed based on the integrated demand response. Firstly, a master–slave game framework of MAIES is established with an energy management agent as leader, an energy operation agent, an energy storage agent, and a user aggregation agent as followers. Secondly, in view of the wind and solar uncertainty, the Monte Carlo method is used to generate random scenarios, and the k-means clustering method and pre-generation elimination technology are used for scenario reduction. Then, according to different flexible characteristics of loads, a multi-load and multi-type integrated demand response model including electric, thermal, and cold energy is built to fully utilize the regulation role of flexible resources. On this basis, the transaction decision-making models of each agent are constructed, and the existence and uniqueness of the Stackelberg equilibrium solution are proved. Finally, the case simulations demonstrate the effectiveness of the proposed optimal scheduling strategy of MAIES. Compared to the scenario without considering the wind and solar uncertainty and the integrated demand response, the rate of renewable energy curtailment was reduced by 6.03% and the carbon emissions of the system were reduced by 1335.22 kg in the scenario considering the proposed method in this paper.

**Keywords:** multi-agent integrated energy system; master–slave game; wind and solar uncertainty; integrated demand response; carbon emission trading; renewable energy

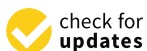



## 1. Introduction

With the increasing demand for energy and increasingly serious environmental pollution, the development of clean, economical, and efficient energy supply is the focus of the energy field [1,2]. The integrated energy system (IES) can fully utilize the complementary characteristics of different energy and can coordinate and optimize the entire process of energy production and consumption. At the same time, it can couple and transform multiple heterogeneous energy flows, and achieve the rational distribution and cascade utilization of energy [3]. Therefore, the IES has become an important form of efficient energy utilization, which plays an important role in promoting the development of new energy and building an environmentally friendly society [4].

As a significant carrier of energy usage, there has been much research on the IES. In [5], an optimal scheduling method of combined power and heat system considering the load demand response and the resident thermal inertia is proposed, which makes full use of the respective characteristics of electric and thermal load. In [6], the heat pumps, combined cooling, heating and power (CCHP) units, and renewable energy are integrated into the energy hub. And the complementarity of various energy sources is used to solve the problem of distribution network congestion. The authors in [7] propose

a multi-objective optimization model of IES based on the carbon trading mechanism and the refined demand response, so as to improve the incomes of the system and promote the accommodation of clean energy. In [8], a distributed solar–biogas residential IES is established, which can supply multiple energy sources in remote locations and make full use of the complementarity between solar energy and biogas. However, the above studies are all about the operation optimization of the IES, without considering the competitive behaviors of interests among different agents in the system. There are complex interest interactions and conflicts among multiple agents in energy production, transmission, and consumption processes, which will bring enormous difficulties and impacts on the operation and regulation of the system. How to ensure the benefits of all parties is a problem that needs to be solved in the optimal scheduling of the IES. In view of these, taking different parts of the system as stakeholders to be involved in the optimal scheduling, a multi-agent integrated energy system (MAIES) is formed.

The operation and optimization of MAIES depend on the cooperation of multiple stakeholders, and the basic problem to be solved is how to describe the interaction between large-scale complex systems and different agents [9]. The optimization of MAIES belongs to the optimization problem of a large-scale complex system, which has a large number of parameters and variables. Because the centralized optimization requires high data transmission, communication, and processing capabilities, and cannot protect the information privacy and security of each agent. Therefore, it is more appropriate to study the distributed optimization of MAIES. The game theory is a method of studying how multiple decision-makers make appropriate decisions based on their information and abilities when there are interests or conflicts between them [10]. In the process of energy transaction, the energy manager first formulates the price strategy according to the energy demand, and then the other agents respond according to the price information. There is a sequential order in the game process between them, which accords with the dynamic game situation of the master–slave hierarchical structure, so the master–slave game model should be used to analyze the interaction between the agents. In [11], the optimal scheduling of cooling, heating, and power supply multi-microgrid systems with electricity interaction is built to minimize the total operating cost of the multi-microgrid system. In [12], a one-master and multi-slave game model between microgrid operators and users with coupled thermoelectric load is established based on the Stackelberg game, which aims to handle the multiparty energy management problem of the grid-connected microgrids. A multi-agent Stackelberg game model is constructed, and the interaction between multiple distributed energy stations and multiple users in IES is studied in [13]. In [14], a modeling and operation method of the IES is proposed, and an integrated model is built based on the Stackelberg game theory to realize a balance of profits between the whole system and the subsystems. In [11–14], although the game interaction of multiple stakeholders is considered, they all focus on optimizing the energy supply side, while ignoring the impact of energy consumption behavior of the user side. With the development of MAIES and the reform of the energy market, the coupling interaction between sources and loads is becoming more obvious, and the traditional vertical integrated structure is transforming into an interactive competitive structure. The energy prices not only affect load demand but also have a reverse effect on energy prices. Therefore, it is of great significance to study the influence of the demand response behavior of users on the optimal scheduling of the system.

Integrated demand response (IDR) can effectively promote the adjustment of energy consumption behaviors of users, which is an important way to realize flexible interaction between the energy demand side and supply side [15]. Most of the existing research is about a single type of demand response behavior [16,17], or multi-type demand response behavior in a single IES [18,19]. In [20,21], the demand response method based on time-sharing electricity price is applied to the game of multi-agent energy system, which can maximize consumer surplus and enable users to obtain a good energy consumption experience. And in [22], the demand response model is constructed based on the transferability and interruptibility of loads in the multi-agent system to give full play to the characteristics

of different types of loads. In [23], taking the peak cutting compensation price issued by the comprehensive energy operator as the leader strategy, and the user's translational load, reducible load, and variable thermal load as the follower strategy, an interactive optimization operation method with one master and multiple slaves is constructed. However, in [20–23], they do not fully consider the characteristics of different types of load and do not adopt diversified and refined demand response methods, and the modeling methods of demand response are relatively simple. In addition, although the above research studies the interest competition behavior of various subjects under the background of IES and considers renewable energy such as wind energy and solar energy, the influence of the uncertainty of wind and solar power output on the optimal operation of IES is ignored. The uncertainty brought by large-scale renewable energy will cause adverse effects on system operation. Therefore, it is necessary to describe the uncertainty of renewable energy output and its impact. Furthermore, carbon emission trading is an effective measure to achieve the goals of carbon peak and carbon neutrality [24]. So, in the game interaction among multiple stakeholders, it is also essential to consider the impact of carbon emissions, so as to ensure the interests of different stakeholders and the overall environmental benefits, and achieve a win-win situation for the economy and environmental protection [25].

For the above problems, this paper proposes a master–slave game optimal operation strategy of MAIES considering the uncertainty of wind and photovoltaic power output and the integrated demand response. The main contributions are as follows:

(1) In view of the randomness of distributed energy generation, the Weibull distribution is used to simulate wind power generation, and the Beta distribution is used to simulate photovoltaic power generation. Then, the Monte Carlo method is adopted to generate random scenarios, and the k-means clustering method and pre-generation elimination technology are adopted to reduce the scenarios, which aims to promote the utilization of renewable energy and improve the economy of system operation.

(2) According to different scheduling potentials and flexible characteristics of loads, an integrated demand response model is established. It includes the electric load demand response model based on electricity price, the thermal load demand response model based on economic incentive, and the cold load demand response model based on fuzzy comfort, so as to give full play to the regulation potential of various loads and improve the enthusiasm of the demand side to participate in the flexible interaction of the system.

(3) Based on Stackelberg game theory, a low-carbon interaction mechanism is established with the energy management agent as a leader, energy operation agent, energy storage agent, and user aggregation agent as followers. According to the roles and benefits of each agent, a two-level game model of one master and multiple slaves with the goal of maximizing the revenue of each agent is established. The simulation results verify the effectiveness and superiority of the strategy proposed in this paper.

This paper is organized as follows. The master–slave game framework of MAIES is introduced and the mathematical models are built in Section 2. Section 3 establishes the decision-making models of each agent of MAIES. The case simulations are presented in Section 4. Finally, the conclusions and future works are drawn in Section 5.

## 2. Framework and Modeling of MAIES

### 2.1. Master–Slave Game Framework of MAIES

The master–slave game framework of MAIES is shown in Figure 1. The energy management agent (EMA), energy operation agent (EOA), energy storage agent (ESA), and user aggregation agent (UAA) are all rational and independent individuals who can make their own decisions and fully participate in the market competition to maximize their own benefits. The EMA can be regarded as an energy agent with two-way energy flow and is the leader in the MAIES energy market. The EMA sets prices for purchasing and selling energy with the goal of maximizing revenue. When the electricity purchased by EMA from EOA cannot satisfy the demands of users, it is necessary for EMA to purchase electricity from the power grid and pay for the carbon emission cost generated by the purchased electricity.

The role of EOA is to provide users with the required electric, thermal, and cold power. The EOA takes the CCHP unit as the core and takes the maximum revenue of energy sales as the objective function to optimize the output of the device. Based on price information, the ESA optimizes its own energy storage and release power between EMA and UAA through low-price energy storage and high-price energy release, thus achieving profitability. The role of UAA is to provide users with energy consumption strategies. According to the price signal, the UAA measures its own revenue and participates in the IDR, so as to adjust energy demand and maximize the comprehensive benefit of the user side. The adjusted actual energy demand will also in turn affect the self-benefits of various stakeholders.

The EMA is the leader, and the EOA, ESA, and UAA are the followers. The energy interaction process of MAIES is divided into two stages in this paper. In the first stage, the EMA sets the energy sales price for UAA and ESA and sets the energy purchase price for EOA. In the second stage, according to the price signal of EMA, the EOA changes the output power of the devices, ESA changes its energy storage and release power and UAA changes its energy consumption strategy. The two stages proceed sequentially, influence each other, and iterate in a loop until equilibrium is reached.

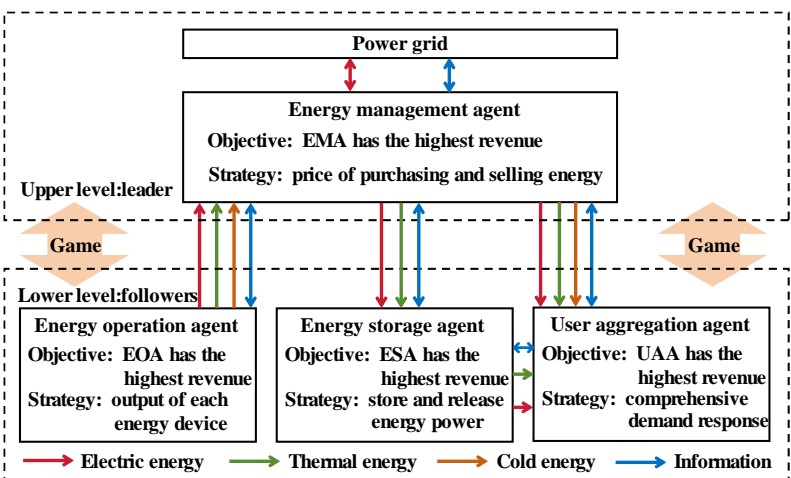

**Figure 1.** Master–slave game framework of MAIES.

### 2.2. Structure and Modeling of CCHP Unit

The CCHP integrates the advantages of renewable energy generation and traditional fuel power generation. It is based on the principle of energy cascade utilization while meeting the different energy demands of users for electric, thermal, and cold energy. The structure of CCHP is shown in Figure 2. The system consists of a wind turbine (WT), photovoltaic (PV), micro gas turbine (MT), waste heat boiler (WHB), gas boiler (GB), heat exchanger (HE), absorption refrigerator (AR), and ice-storage air-conditioners (ISAC).

The fuel cost of MT and GB can be expressed as:

$$C_t^{fu} = a^{MT}\left(P_t^{MT}\right)^2 + b^{MT}P_t^{MT} + c^{MT} \\ + a^{GB}\left(H_t^{GB}\right)^2 + b^{GB}H_t^{GB} + c^{GB} \tag{1}$$

where $P_t^{MT}$ and $H_t^{GB}$ are the electric power generated by MT and the thermal power generated by GB at time $t$, respectively. $a^{MT}$, $b^{MT}$, $c^{MT}$ and $a^{GB}$, $b^{GB}$, $c^{GB}$ are the fuel coefficients of MT and GB, respectively.

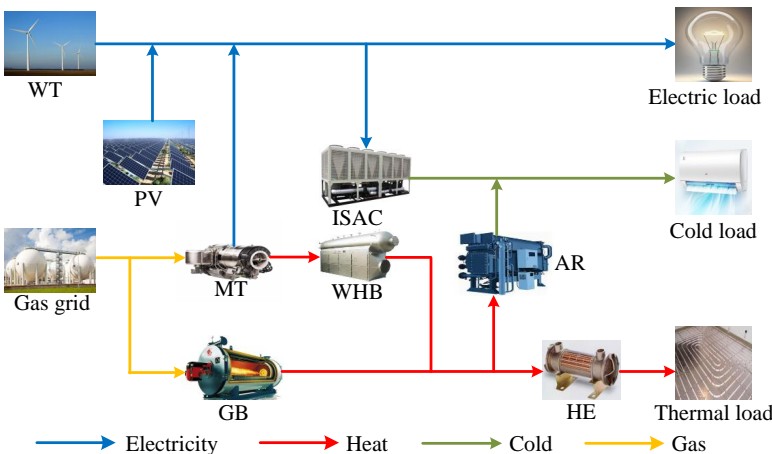

**Figure 2.** Structure of CCHP.

The waste heat generated by MT is absorbed by WHB and converted into thermal energy, which is combined with the thermal energy generated by GB. It generates thermal energy through HE and cold energy through AR. The model is represented as follows:

$$H_t^{MT} = \frac{1 - \eta^{MT} - \eta^{loss}}{\eta^{MT}} P_t^{MT} \tag{2}$$

$$H_t^{WHB} = H_t^{MT} \eta^{WHB} \tag{3}$$

$$H_t^{HE} = \left(H_t^{WHB} + H_t^{GB}\right) \eta^{HE} \tag{4}$$

$$C_t^{AR} = \left(H_t^{WHB} + H_t^{GB}\right) \eta^{AR} \tag{5}$$

where $H_t^{MT}$ is the waste thermal power generated by MT, and $H_t^{WHB}$ is the thermal power recovered by WHB. $H_t^{HE}$ and $C_t^{AR}$ are the thermal and cold power generated by HE and AR, respectively. $\eta^{MT}$ and $\eta^{loss}$ are the electricity generation efficiency and heat loss coefficient of MT, respectively. $\eta^{WHB}$ is the heat efficiency of WHB. $\eta^{HE}$ and $\eta^{AR}$ are the heating and cooling efficiency of HE and AR, respectively.

In addition, the MT and GB need to satisfy the upper and lower limit constraints of output power and climbing rate.

$$P^{MT,\min} \leq P_t^{MT} \leq P^{MT,\max} \tag{6}$$

$$P^{MT,down} \leq P_t^{MT} - P_{t-1}^{MT} \leq P^{MT,up} \tag{7}$$

$$H^{GB,\min} \leq H_t^{GB} \leq H^{GB,\max} \tag{8}$$

$$H^{GB,down} \leq H_t^{GB} - H_{t-1}^{GB} \leq H^{GB,up} \tag{9}$$

where $P^{MT,\min}$ and $P^{MT,\max}$ are the minimum and maximum electric power of MT, respectively. $P^{MT,up}$ and $P^{MT,down}$ are the upper and lower limits of the climbing rate of MT, respectively. $H^{GB,\min}$ and $H^{GB,\max}$ are the minimum and maximum thermal power of GB, respectively. $H^{GB,up}$ and $H^{GB,down}$ are the upper and lower limits of climbing rate of GB, respectively.

### 2.3. Carbon Trading Model

Carbon trading is a trading mechanism to control carbon emissions. Under the carbon trading mechanism, carbon emissions become freely traded commodities. When the carbon emissions of the system exceed the carbon quota, the corresponding carbon trading cost will be paid. When the carbon emissions are lower than the carbon quota, the remaining carbon quota can be sold in the carbon trading market [26].

For the power industry, the allocation of initial carbon emissions quota is generally based on free of charge. In this paper, the energy purchased by the system from the power grid is generated by thermal power units, so the main carbon emission sources of the system are MT, GB, and purchased energy. The model of the initial carbon quota is [7]:

$$\begin{cases} E^{F,grid} = \gamma_p P_t^{grid} \\ E^{F,MT} = \gamma_h(H_t^{MT} + \delta^{e,h} P_t^{MT}) \\ E^{F,GB} = \gamma_h H_t^{GB} \end{cases} \tag{10}$$

where $E^{F,MT}$, $E^{F,GB}$, and $E^{F,grid}$ are the initial carbon quotas of MT, GB, and electricity purchased from the power grid, respectively. $P_t^{grid}$ is the electric power purchased from the grid. $\gamma_p$ and $\gamma_h$ are the carbon emission quotas per unit of electricity supply and heat supply, respectively. $\delta^{e,h}$ is the transformation coefficient from electricity to heat.

The actual carbon emission model can be represented as:

$$\begin{cases} E^{A,grid} = \sigma_e P_t^{grid} \\ E^{A,MT} = \sigma_g P_t^{MT}/\eta_{MT}\lambda_{gas} \\ E^{A,GB} = \sigma_g H_t^{GB}/\eta_{GB}\lambda_{gas} \end{cases} \tag{11}$$

where $E^{A,MT}$, $E^{A,GB}$, and $E^{A,grid}$ are the actual carbon emissions of MT, GB, and energy purchased from the grid, respectively. $\lambda_{gas}$ is the calorific value of natural gas. $\sigma_e$ and $\sigma_g$ are the carbon emission coefficients of unit electricity consumption and unit natural gas consumption, respectively. Therefore, the carbon trading cost of purchasing electricity from the power grid is:

$$C^{grid,c} = \varepsilon_c \left( E^{A,grid} - E^{F,grid} \right) \tag{12}$$

The carbon trading cost of MT and GB is:

$$C^{MG,c} = \varepsilon_c \left( E^{A,MT} + E^{A,GB} - E^{F,MT} - E^{F,GB} \right) \tag{13}$$

where $\varepsilon_c$ is the carbon trading price. $C^{MG,c}$ is the equivalent total carbon trading cost of MT and GB.

### 2.4. The Model of Wind and Solar Uncertainty

The output models of wind turbines and photovoltaic generators are:

$$P_t^{WT} = \begin{cases} 0, v_t < v_{CI} \, or \, v_t > v_{CO} \\ P_R \frac{v_t - v_{CI}}{v_R - v_{CI}}, v_{CI} \le v_t < v_R \\ P_R, v_R \le v_t < v_{CO} \end{cases} \tag{14}$$

$$P_t^{PV} = P_{PVR} \frac{G_t}{G_R} \{1 + \tau[T_t - T_R]\} \tag{15}$$

where $P_t^{WT}$ and $P_t^{PV}$ are the WT and PV output power, respectively. $P_R$ and $P_{PVR}$ are the rated WT and PV output power, respectively. $v_t$ is the wind speed. $v_{CI}$, $v_{CO}$, and $v_R$ are the cut-in wind speed, cut-out wind speed, and rated wind speed, respectively. $G_R$ is the rated light radiation, and $G_t$ is the light radiation. $T_R$ is the rated temperature, and $T_t$ is the temperature.

In view of the randomness of wind and solar, existing studies have proved that wind speed obeys Weibull distribution [27] and light intensity obeys Beta distribution [28]. The probability distributions are:

$$f[v_t] = \frac{k_s}{c_s} \left(\frac{v_t}{c_s}\right)^{k_s-1} \exp\left\{-\left(\frac{v_t}{c_s}\right)^{k_s}\right\} \qquad (16)$$

$$f\left[\frac{P_t^{PV}}{P_{PVR}}\right] = \frac{1}{B(\alpha_S, \beta_S)} \left[\frac{P_t^{PV}}{P_{PVR}}\right]^{\alpha_S-1} \left[1 - \frac{P_t^{PV}}{P_{PVR}}\right]^{\beta_S-1} \qquad (17)$$

where $c_s$ and $k_s$ are the proportion parameter and shape parameter of the Weibull distribution, respectively. $\alpha_s$ and $\beta_s$ are the shape parameters of Beta distribution.

Based on the probability distribution and the historical data of wind speed and light, the Monte Carlo (MC) random sampling algorithm is used to generate multiple scenes, and the k-means clustering method is adopted to reduce the scenes to obtain a small number of typical scenes with different probabilities. Then, the pre-generation elimination method is used to further reduce the scenes. Finally, Y typical scenarios and their corresponding probabilities are obtained. The system is analyzed by calculating the revenue of each stakeholder in each scenario and multiplying the corresponding probability and accumulating them.

### 2.5. IDR Model

The users have various energy needs of electric, thermal, and cold energy in MAIES. The IDR considered in this paper includes the electric load demand response based price, thermal load demand response based incentive, and cold load demand response based on fuzzy comfort.

#### 2.5.1. Model of Electric Load Demand Response

In economics, the price elasticity of demand is commonly used to reflect the response of demand to price changes, while the electric load demand response model based on the electricity price elasticity matrix can reflect the sensitivity of users to electricity price at a certain time and adjacent time.

$$\delta = \frac{\alpha}{\Delta\alpha} \frac{\Delta P}{P} \qquad (18)$$

$$\mathbf{P}^{UAA,IDR} = \begin{bmatrix} P_1^0 \\ P_2^0 \\ \vdots \\ P_t^0 \end{bmatrix} + \\ \begin{bmatrix} P_1^0 & 0 & \cdots & 0 \\ 0 & P_2^0 & \cdots & 0 \\ \vdots & \vdots & \ddots & \vdots \\ 0 & 0 & \cdots & P_t^0 \end{bmatrix} \begin{bmatrix} \delta_{11} & \delta_{12} & \cdots & \delta_{1t} \\ \delta_{21} & \delta_{22} & \cdots & \delta_{2t} \\ \vdots & \vdots & \ddots & \vdots \\ \delta_{t1} & \delta_{t2} & \cdots & \delta_{tt} \end{bmatrix} \begin{bmatrix} \frac{\Delta\alpha_1}{\alpha_1} \\ \frac{\Delta\alpha_2}{\alpha_2} \\ \vdots \\ \frac{\Delta\alpha_t}{\alpha_t} \end{bmatrix} \qquad (19)$$

$$\mathbf{P}^{UAA,IDR} = \left(1 + \delta_{tt}\frac{\Delta\alpha_t}{\alpha_t} + \sum_{n=1,n\neq t}^{T} \delta_{tn}\frac{\Delta\alpha_t}{\alpha_t}\right)\mathbf{P}^0 \qquad (20)$$

where $\delta$, $\delta_{tt}$ and $\delta_{tn}$ are elastic coefficient, self-elastic coefficient, and cross-elastic coefficient, respectively, which are used to measure the dependence of consumer electricity consumption on the increase in electric price at a certain time and adjacent time [29]. The values of $\delta_{tt}$ and $\delta_{tn}$ are −0.2 and 0.03, respectively. $\alpha_t$ and $\Delta\alpha_t$ are the electric price and the

change of electric price, respectively. $\mathbf{P}^{UAA,IDR} = \left[ P_1^{UAA,IDR}, P_2^{UAA,IDR}, \cdots P_t^{UAA,IDR} \right]^T$,
$\mathbf{P}^0 = \left[ P_1^0, P_2^0, \cdots P_t^0 \right]^T$. $P_t^{UAA,IDR}$ and $P_t^0$ are the electric load before and after the IDR.

### 2.5.2. Model of Thermal Load Demand Response

The thermal load demand response based on incentive is used to transfer and reduce the thermal load of the user. The thermal load is divided into basic thermal load, transferable thermal load, and reducible thermal load. The basic thermal load requires high reliability and a fixed energy consumption time to ensure normal production and life of users. The transferable thermal load refers to the load for which the energy consumption time can be flexibly adjusted, and the total amount of transferable thermal load remains unchanged during the scheduling cycle. The reducible thermal load refers to the load that can withstand a certain interruption, power reduction, or operating time reduction. In order to improve the enthusiasm of the users to adjust their own energy consumption behaviors, the model of thermal load demand response is established based on the economic incentive mechanism. Specifically, the economic reward is given to users according to load transfer and reduction. The model of transferable thermal load is:

$$H_{t_{start}+n} = U_{t_{start}+n}^H H_{n+1}^{tran} \quad n \in [0, t_{last}] \tag{21}$$

$$\sum_{t=1}^{T} U_t^H = \sum_{t=t_{start}}^{t_{end}} U_t^H = t_{last} \tag{22}$$

$$U_t^H - U_{t-1}^H \leq B_t^H \tag{23}$$

where $H_t^{tran}$ is the transferable thermal load. $[t_{start}, t_{end}]$ is the translation range of transferable thermal load. $t_{last}$ is the duration of the load. $U_t^H$ and $B_t^H$ are the 0–1 variable of the operation state and start-stop state of the load, respectively. The model of reducible thermal load is:

$$\sum_{t=t_{from}}^{t=t_{to}} H_t^{redu} = H_{sum}^{redu} \tag{24}$$

$$H_{min}^{redu} \leq H_t^{redu} \leq H_{max}^{redu} \tag{25}$$

where $t_{from}$ and $t_{to}$ are the start and stop time of reducible thermal load, respectively. $H_{max}^{redu}$ and $H_{min}^{redu}$ are the upper and lower limits of reducible thermal load. $H_t^{redu}$ and $H_{sum}^{redu}$ are the actual reducible thermal load and the total amount of load reduction.

$$H_t^{UAA,IDR} = H_t^{base} + H_t^{tran} - H_t^{redu} \tag{26}$$

$$C_t^{h,IDR} = \sum_{t=1}^{T} \left( H_t^{tran} c_t^{h,tran} + H_t^{redu} c_t^{h,redu} \right) \tag{27}$$

where $H_t^{base}$ and $H_t^{UAA,IDR}$ are the basic thermal load and the thermal load after the IDR, respectively. $c_t^{h,tran}$ and $c_t^{h,redu}$ are the compensation for unit thermal load translation and reduction, respectively. $C_t^{h,IDR}$ is the compensation benefits of participating in the thermal load demand response.

### 2.5.3. Model of Cold Load Demand Response

The hot and cold comfort of the human is affected by many factors, and it is a fuzzy random variable, so the evaluation of human comfort is a fuzzy concept. It is difficult for humans to detect temperature changes within a certain range [29], and comfort is affected when the temperature is adjusted in this range. The cold load is regarded as a flexible adjustable load to be involved in the cold demand response based on fuzzy comfort. In this

paper, the predicted mean vote (PMV) index of cold and hot feelings is used to represent the user's feeling of ambient temperature [30], and its expression is:

$$
\begin{aligned}
K^{PMV} = {}& \left(0.303e^{-0.036D} + 0.0275\right)\big[D(1-\eta) \\
& -3.054(5.765 - 0.007H - P_a) - 0.42(H - 58.15) \\
& -0.0173D(5.867 - P_a) - 0.0014D(34 - T_a) \\
& -3.9 \times 10^{-8} K_{cl}\left(T_{cl}^4 - T_{mrt}^4\right) - K_{cl}h_{cl}(t_{cl} - T_a)\big]
\end{aligned}
\tag{28}
$$

where $D$ is the metabolic rate. $e$ is a constant. $\eta$ is the heat dissipation rate of the human body. $t_{cl}$ is the average surface temperature of the human body. $T_{mrt}$ is the average radiation temperature of the environment. According to reference [30], the optimal comfort of the human body corresponds to the temperature of 24 °C, and $K^{PMV} \in [-0.9, 0.9]$. The relationship between $K^{PMV}$ and temperature $T$ is:

$$
K^{PMV} = \begin{cases} 0.3895(T-24), T \geq 24 \\ 0.4065(-T+24), T < 24 \end{cases}
\tag{29}
$$

$$
C_t^{UAA,IDR} = S\mu\left(T_t^{out} - T_t^{in}\right) + (CS/\Delta t)\left(T_t^{in} - T_{t-1}^{in}\right)
\tag{30}
$$

$$
T^{\min} \leq T_t^{in} \leq T^{\max}
\tag{31}
$$

where $C_t^{UAA,IDR}$ is the cold load. $S$ is the cooling area. $\mu$ is the heat loss per unit cooling area under unit temperature difference, the value is $1.037 \times 10^4$ J/(m²· °C). $C$ is the heat capacity per unit cooling area, the value is $1.63 \times 10^5$ J/(m²· °C). $T_t^{in}$ and $T_t^{out}$ are the indoor and outdoor temperature, respectively.

## 3. Decision-Making Models of Each Agent

### 3.1. EMA Model

The EMA sets energy purchase and sale prices, guides the output of various devices, and encourages users to adjust their energy usage behavior by IDR. The objective function of EMA is:

$$
\begin{aligned}
\max F^{EMA} = \sum_{\omega=1}^{\Omega} \Bigg\{ \sum_{t=1}^{T} \Big[ X_\omega \Big( & C_{t,\omega}^{EMA,u} + C_{t,\omega}^{EMA,s} \\
& - C_{t,\omega}^{EMA,buy} - C_{t,\omega}^{EMA,grid} - C_{t,\omega}^{grid,c} - C_{t,\omega}^{h,IDR} \Big) \Big] \Bigg\}
\end{aligned}
\tag{32}
$$

where $F^{EMA}$ is the total daily revenue of EMA. $\Omega$ is the total number of scenarios. $X_\omega$ is the probability of scenario $\Omega$ occurring. $T$ is the total optimal scheduling cycle. $C_{t,\omega}^{EMA,u}$ and $C_{t,\omega}^{EMA,s}$ are the energy sales revenue obtained from UAA and ESA in scenario $\omega$ at time $t$, respectively. $C_{t,\omega}^{EMA,buy}$ is the cost for EMA to purchase energy from EOA, and $C_{t,\omega}^{EMA,grid}$ is the electricity exchange cost between EMA and power grid. $C_{t,\omega}^{grid,c}$ is the carbon trading cost of EMA to purchase energy from the power grid, which can be calculated by Equation (12).

$$
C_{t,\omega}^{EMA,u} = \left(k_{t,\omega}^{s,e} P_{t,\omega}^{UAA,IDR} + k_{t,\omega}^{s,h} H_{t,\omega}^{UAA,IDR} + k_{t,\omega}^{s,c} C_{t,\omega}^{UAA,IDR}\right)\Delta t
\tag{33}
$$

$$
C_{t,\omega}^{EMA,s} = \left(k_{t,\omega}^{s,e} P_{t,\omega}^{ES,ch} + k_{t,\omega}^{s,h} H_{t,\omega}^{HS,ch}\right)\Delta t
\tag{34}
$$

$$
C_{t,\omega}^{EMA,buy} = \left(r_{t,\omega}^{b,e} P_{t,\omega}^{EOA,e} + r_{t,\omega}^{b,h} H_{t,\omega}^{EOA,h} + r_{t,\omega}^{b,c} C_{t,\omega}^{EOA,c}\right)\Delta t
\tag{35}
$$

$$
\begin{aligned}
C_{t,\omega}^{EMA,grid} = \Big[ &\max\Big( P_{t,\omega}^{UAA,IDR} - P_{t,\omega}^{EOA,e}, 0 \Big) c_t^{s,grid} \\
&+ \min\Big( P_{t,\omega}^{UAA,IDR} - P_{t,\omega}^{EOA,e}, 0 \Big) c_t^{b,grid} \Big] \Delta t
\end{aligned}
\tag{36}
$$

where $P_{t,\omega}^{ES,ch}$ and $H_{t,\omega}^{HS,ch}$ are the electric and thermal power sold by EMA to ESA, respectively. $P_{t,\omega}^{EOA,e}$, $H_{t,\omega}^{EOA,h}$ and $C_{t,\omega}^{EOA,c}$ are the electric, thermal and cold power purchased by EMA from EOA. $k_{t,\omega}^{s,e}$, $k_{t,\omega}^{s,h}$ and $k_{t,\omega}^{s,c}$ are the prices of electric, thermal, and cold energy sold by EMA to UAA, respectively. $r_{t,\omega}^{b,e}$, $r_t^{b,h,\omega}$ and $r_t^{b,c,\omega}$ are the prices for EMA to purchase electric, thermal and cold energy from EOA, respectively. $c_t^{s,grid}$ and $c_t^{b,grid}$ are the selling and purchasing prices of electricity from EMA to the power grid, respectively.

In order to ensure the interests of all parties, the purchase and sale prices of energy for EMA must meet the following constraints:

$$
\begin{cases}
c_t^{b,grid} < k_{t,\omega}^{s,e} < c_t^{s,grid} \\
c_t^{b,grid} < r_{t,\omega}^{b,e} < c_t^{s,grid}
\end{cases}
\tag{37}
$$

$$
\begin{cases}
k_t^{s,h,\min} < k_{t,\omega}^{s,h} < k_t^{s,h,\max}, r_t^{b,h,\min} < r_{t,\omega}^{b,h} < r_t^{b,h,\max} \\
k_t^{s,c,\min} < k_{t,\omega}^{s,c} < k_t^{s,c,\max}, r_t^{b,c,\min} < r_{t,\omega}^{b,c} < r_t^{b,c,\max}
\end{cases}
\tag{38}
$$

$$
\begin{cases}
\displaystyle\sum_{t=1}^{T} k_{t,\omega}^{s,e} \le T\bar{k}_t^{s,e,\max} \\
\displaystyle\sum_{t=1}^{T} k_{t,\omega}^{s,h} \le T\bar{k}_t^{s,h,\max} \\
\displaystyle\sum_{t=1}^{T} k_{t,\omega}^{s,c} \le T\bar{k}_t^{s,c,\max}
\end{cases}
\tag{39}
$$

where $k_t^{s,h,\min}$, $r_t^{b,h,\min}$ and $k_t^{s,h,\max}$, $r_t^{b,h,\max}$ are the lowest and highest prices of thermal and cold energy, respectively. $\bar{k}_t^{s,e,\max}$, $\bar{k}_t^{s,h,\max}$ and $\bar{k}_t^{s,c,\max}$ are the average selling electric, heat and cold prices, respectively.

*3.2. EOA Model*

The optimization goal of EOA is to maximize its own revenues, that is, the difference between energy sales revenue and device operating costs. The objective function is:

$$
\max F^{EOA} = \sum_{\omega=1}^{\Omega} \left\{ \sum_{t=1}^{T} \Big[ X_\omega \Big( C_{t,\omega}^{EOA,sell} - C_{t,\omega}^{fu} - C_{t,\omega}^{EOA,om} - C_{t,\omega}^{MG,c} \Big) \Big] \right\}
\tag{40}
$$

where $F^{EOA}$ is the total revenue of EOA. $C_{t,\omega}^{EOA,sell}$ is the energy sales revenue of EOA. $C_{t,\omega}^{EOA,om}$ is the cost of device operation and maintenance. $C_{t,\omega}^{MG,c}$ is the carbon trading cost of EOA, which can be calculated by Equation (13). The cost of device operation and maintenance can be calculated by:

$$
C_{t,\omega}^{EOA,om} = \sum_{t=1}^{T} \sum_{k=1}^{7} \phi_k P_{t,\omega}^k
\tag{41}
$$

where the value of *k* is 1,2,..., 8, which represent WT, PV, MT, GB, WHB, HE, AR, and ISAC, respectively. $P_{t,\omega}^k$ is the output power of device *k*. $\phi_k$ is the cost coefficient for the operation and maintenance of device *k*.

The electric, thermal, and cold power output by EOA need to meet the following expressions:

$$
P_{t,\omega}^{EOA,e} = P_{t,\omega}^{WT} + P_{t,\omega}^{PV} + P_{t,\omega}^{MT} - P_{t,\omega}^{ISAC}
\tag{42}
$$

$$
H_{t,\omega}^{EOA,h} = H_{t,\omega}^{HE} = \Big( H_{t,\omega}^{WHB} + H_{t,\omega}^{GB} \Big) \eta^{HE}
\tag{43}
$$

$$C_{t,\omega}^{EOA,c} = C_{t,\omega}^{AR} + C_{t,\omega}^{IC} + C_{t,\omega}^{IM} \tag{44}$$

where $P_{t,\omega}^{ISAC}$ is the electric power consumption of ISAC. $C_{t,\omega}^{IC}$ and $C_{t,\omega}^{IM}$ are the output cooling power and ice melting power of ISAC, respectively [31].

*3.3. ESA Model*

At present, there are some problems in user-side distributed energy storage, such as high investment cost, long cost recovery period, and low safety and reliability, which limit the wide use of user-side distributed energy storage [32]. Therefore, this paper proposes to establish a centralized energy storage station, which can not only reduce the storage cost per unit of power but also avoid the security problems caused by improper maintenance of distributed energy storage. The ESA makes revenues by storing energy at low prices and selling energy at high prices between EMA and UAA, and the devices contain ES and HS. As an independent energy storage agent, the auxiliary service is a part of the revenue, and this revenue has been considered in the published comprehensive energy sales price. The objective function of ESA is:

$$\max F^{ESA} = \sum_{\omega=1}^{\Omega} \left\{ \sum_{t=1}^{T} \left[ X_\omega \left( C_{t,\omega}^{ESA,dis} - C_{t,\omega}^{EMA,s} - C_{t,\omega}^{ESA,om} \right) \right] \right\} \tag{45}$$

$$C_{t,\omega}^{ESA,dis} = \left( \lambda_{t,\omega}^{ES} P_{t,\omega}^{ES,dis} + \lambda_{t,\omega}^{HS} H_{t,\omega}^{HS,dis} \right) \Delta t \tag{46}$$

$$C_{t,\omega}^{ESA,om} = \left[ \varepsilon^{ES,om} \left( P_{t,\omega}^{ES,ch} + P_{t,\omega}^{ES,dis} \right) + \varepsilon^{HS,om} \left( H_{t,\omega}^{HS,ch} + H_{t,\omega}^{HS,dis} \right) \right] \Delta t \tag{47}$$

$$\begin{cases} \lambda_{t,\omega}^{ES} = \lambda_{t,\omega}^{ES,sell} + \tau^e \\ \lambda_{t,\omega}^{HS} = \lambda_{t,\omega}^{HS,sell} + \tau^h \end{cases} \tag{48}$$

where $F^{ESA}$ is the total revenue of ESA. $C_{t,\omega}^{ESA,dis}$ is the energy release revenue of ESA. $C_t^{ESA,om,\omega}$ is the operation and maintenance cost of ESA. $\lambda_{t,\omega}^{ES}$ and $\lambda_{t,\omega}^{HS}$ are the comprehensive prices at which the ESA sells electric and thermal energy, respectively. $\varepsilon^{ES,om}$ and $\varepsilon^{HS,om}$ are the operation and maintenance cost coefficients of ES and HS, respectively. $\lambda_{t,\omega}^{ES,sell}$ and $\lambda_{t,\omega}^{HS,sell}$ are the basic selling prices of electric and thermal energy of EAS, respectively. $\tau^e$ and $\tau^h$ are the unit revenue of ESA participating in auxiliary services.

The operational constraints that energy storage devices need to meet are:

$$E_{t,\omega}^x = E_{t-1,\omega}^x (1 - \gamma^x) + \left( P_{t,\omega}^{x,ch} \eta^{x,ch} - \frac{P_{t,\omega}^{x,dis}}{\eta^{x,dis}} \right) \Delta t \tag{49}$$

$$E^{x,\min} \le E_{t,\omega}^x \le E^{x,\max} \tag{50}$$

$$E_{0,\omega}^x = E_{24,\omega}^x \tag{51}$$

$$0 \le P_{t,\omega}^{x,ch} \le P^{\max,x,ch} B_{t,\omega}^{x,ch} \tag{52}$$

$$0 \le P_{t,\omega}^{x,dis} \le P^{\max,x,dis} B_{t,\omega}^{x,dis} \tag{53}$$

$$0 \le B_{t,\omega}^{x,ch} + B_{t,\omega}^{x,dis} \le 1 \tag{54}$$

where $x \in \{ES, HS\}$, $E_{t,\omega}^x$ is the storage energy of device $x$. $\eta^{x,ch}$ and $\eta^{x,dis}$ are the efficiency of energy storage and release of device $x$. $E^{x,\min}$ and $E^{x,\max}$ are the minimum and maximum

values of $E_{t,\omega}^{x}$, respectively. $\gamma^{x}$ is the energy self-loss rate of the energy storage device. $P^{\max,x,ch}$ and $P^{\max,x,dis}$ are the maximum power of energy storage and release of device $x$, respectively. $B_{t,\omega}^{x,ch}$ and $B_{t,\omega}^{x,dis}$ are the 0–1 variables of the energy storage and release behaviors of device $x$, respectively.

*3.4. UAA Model*

In this paper, the benefit obtained by UAA is defined as the sum of the satisfaction obtained by users purchasing electric, thermal, and cold energy, and the quadratic function is used to express the benefit obtained from energy consumption.

$$
\begin{aligned}
U_{t,\omega}^{UAA} = {} & \zeta_e P_{t,\omega}^{UAA,IDR} - \frac{\psi_e}{2} \left( P_{t,\omega}^{UAA,IDR} \right)^2 + \zeta_h H_{t,\omega}^{UAA,IDR} \\
& - \frac{\psi_h}{2} \left( H_{t,\omega}^{UAA,IDR} \right)^2 + \zeta_c C_{t,\omega}^{UAA,IDR} - \frac{\psi_c}{2} \left( C_{t,\omega}^{UAA,IDR} \right)^2
\end{aligned}
\tag{55}
$$

where $\zeta_e$, $\psi_e$, $\zeta_h$, $\psi_h$ and $\zeta_c$, $\psi_c$ are the coefficient of users' preference for electric, thermal and cold energy.

Based on the given energy prices, the UAA optimizes its own load power by participating in IDR. The objective function of UAA is the difference between the user's benefit function and the energy consumption cost, expressed as:

$$
\max F^{UAA} = \sum_{\omega=1}^{\Omega} \left\{ \sum_{t=1}^{T} \left[ X_{\omega} \left( U_{t,\omega}^{UAA} + C_{t,\omega}^{h,IDR} - C_{t,\omega}^{EMA,u} - C_{t,\omega}^{ESA,dis} \right) \right] \right\}
\tag{56}
$$

*3.5. Master–Slave Game Model*

The master–slave game model can be expressed as:

$$
G = \left\{ R; \delta_{\omega}^{EMA}; \varphi_{\omega}^{EOA}; \varphi_{\omega}^{ESA}; \varphi_{\omega}^{UAA}; F^{EMA}; F^{EOA}; F^{ESA}; F^{UAA} \right\}
\tag{57}
$$

(1) Participants: 1 EMA, 1 EOA, 1 ESA, and 1 UAA, the set of participants can be represented as $R = \{EMA, EOA, ESA, UAA\}$.

(2) Strategies: The strategy of leader EMA is the purchase and sale prices of energy, which can be expressed as the vector $\delta_{\omega}^{EMA} = \left\{ \mathbf{k}_{t,\omega}^{s,e}, \mathbf{k}_{t,\omega}^{s,h}, \mathbf{k}_{t,\omega}^{s,c}, \mathbf{r}_{t,\omega}^{b,e}, \mathbf{r}_{t,\omega}^{b,h}, \mathbf{r}_{t,\omega}^{b,c} \right\}$. The strategy of follower EOA is the output power of the energy supply devices in EOA, which can be expressed as $\varphi_{\omega}^{EOA} = \left\{ \mathbf{P}_{t,\omega}^{MT}, \mathbf{H}_{t,\omega}^{GB} \right\}$. The strategy of follower ESA is the energy storage and release power of ES and HS, which can be expressed as $\varphi_{\omega}^{ESA} = \left\{ \mathbf{P}_{t,\omega}^{ES}, \mathbf{H}_{t,\omega}^{HS} \right\}$. The strategy of follower UAA is the demand response of electric, thermal and cold loads, which can be expressed as $\varphi_{\omega}^{UAA} = \left\{ \mathbf{P}_{\omega}^{UAA,IDR}, \mathbf{H}_{\omega}^{UAA,IDR}, \mathbf{C}_{\omega}^{UAA,IDR} \right\}$.

(3) Revenues: The revenues of each participant are expressed as each objective function, which can be calculated by Formulas (32), (40), (45) and (55).

When all followers make the optimal response according to the leader's strategy and the leader accepts the responses, the game reaches Stackelberg equilibrium. In the Stackelberg equilibrium solution, it is impossible for any participant to gain more benefits by unilaterally changing the strategy.

The proof of the existence and uniqueness of equilibrium solutions in games is as follows:

**Theorem 1.** *When the master–slave game model meets the following conditions, there exists a unique Stackelberg equilibrium solution:*

(1) The revenue function of the game participants is a non-empty and continuous function of the game strategy set.

(2) When the strategy of the leader is determined, all followers have a unique optimal solution.

(3) When the strategy of the follower is determined, the leader has a unique optimal solution.

**Proof.** (1) The strategy of the leader EMA needs to meet the Formulas (37)–(39), the strategy of EOA needs to meet the Formulas (6)–(9), the strategy of ESA needs to meet the Formulas (49)–(54), and the strategy of the UAA needs to meet the Formulas (18)–(31), so the policy set of each participant is non-empty and compact convex.

(2) Proving that when the strategy of the leader is determined, all followers have a unique optimal solution. For ESA, when the leader's energy sales prices $k_{t,\omega}^{s,e}$ and $k_{t,\omega}^{s,h}$ are determined, it can be seen from the objective function (45) that the revenues and decisions of ESA vary linearly. When the constraints (49)–(54) are met, there must be a unique optimal solution for ESA.

For UAA, when $k_{t,\omega}^{s,e}$, $k_{t,\omega}^{s,h}$ and $k_{t,\omega}^{s,c}$ are determined, let formula (56) take the first partial derivative for $P_{t,\omega}^{UAA,IDR}$, $H_{t,\omega}^{UAA,IDR}$ and $C_{t,\omega}^{UAA,IDR}$, respectively:

$$\begin{cases} \dfrac{\partial F^{UAA}}{\partial P_{t,\omega}^{UAA,DR}} = \zeta_e - \psi_e P_{t,\omega}^{UAA,IDR} - k_{t,\omega}^{s,e} \\[2mm] \dfrac{\partial F^{UAA}}{\partial H_{t,\omega}^{UAA,DR}} = \zeta_h - \psi_h H_{t,\omega}^{UAA,IDR} - k_{t,\omega}^{s,h} \\[2mm] \dfrac{\partial F^{UAA}}{\partial C_{t,\omega}^{UAA,DR}} = \zeta_c - \psi_h C_{t,\omega}^{UAA,IDR} - k_{t,\omega}^{s,c} \end{cases} \tag{58}$$

Let the first-order partial derivatives be set to 0, respectively, and the following results can be obtained:

$$\begin{cases} P_{t,\omega}^{UAA,IDR,0} = \dfrac{\zeta_e - k_{t,\omega}^{s,e}}{\psi_e} \\[2mm] H_{t,\omega}^{UAA,IDR,0} = \dfrac{\zeta_h - k_{t,\omega}^{s,h}}{\psi_h} \\[2mm] C_{t,\omega}^{UAA,IDR,0} = \dfrac{\zeta_c - k_{t,\omega}^{s,c}}{\psi_c} \end{cases} \tag{59}$$

Let Formula (56) take the second partial derivative for $P_{t,\omega}^{UAA,IDR}$, $H_{t,\omega}^{UAA,IDR}$ and $C_{t,\omega}^{UAA,IDR}$, respectively, the following results can be obtained:

$$\begin{cases} \dfrac{\partial^2 F^{UAA}}{\partial \left(P_{t,\omega}^{UAA,IDR}\right)^2} = -\psi_e \\[3mm] \dfrac{\partial^2 F^{UAA}}{\partial \left(H_{t,\omega}^{UAA,IDR}\right)^2} = -\psi_h \\[3mm] \dfrac{\partial^2 F^{UAA}}{\partial \left(C_{t,\omega}^{UAA,IDR}\right)^2} = -\psi_h \end{cases} \tag{60}$$

Because the values of $\psi_e$, $\psi_h$, and $\psi_c$ both are positive, the second partial derivatives are all less than zero, and the function has a maximum point. Therefore, when the energy sales prices of EMA are determined, the UAA has the unique optimal solution.

Similarly, for EOA, when $r_{t,\omega}^{b,e}$, $r_{t,\omega}^{b,h}$ and $r_{t,\omega}^{b,c}$ are determined, let Formula (40) take the second partial derivative for $P_{t,\omega}^{MT}$ and $H_{t,\omega}^{GB}$, respectively, the following results can be obtained:

$$\begin{cases} \dfrac{\partial^2 F^{EOA}}{\partial \left(P_{t,\omega}^{MT}\right)^2} = -2a^{MT} \\[3mm] \dfrac{\partial^2 F^{EOA}}{\partial \left(H_{t,\omega}^{GB}\right)^2} = -2a^{GB} \end{cases} \tag{61}$$

Due to $a^{MT}$ and $a^{GB}$ being positive, the function has a maximum point, i.e., the unique optimal solution of EOA is available.

(3) Proving that when the follower strategy is determined, the leader has a unique optimal solution. Assuming that EMA needs to purchase electricity from the power grid

to satisfy the energy requirement of UAA, and substituting the unique optimal solution of the followers obtained from the above proof into Equation (32). So, the second partial derivatives of Equation (32) about $k_{t,\omega}^{s,e}$, $k_{t,\omega}^{s,h}$, $k_{t,\omega}^{s,c}$, $r_{t,\omega}^{b,e}$, $r_{t,\omega}^{b,h}$ and $r_{t,\omega}^{b,c}$ are:

$$\begin{cases} \dfrac{\partial^2 F^{EMA}}{\partial (k_{t,\omega}^{s,e})^2} = -\dfrac{2}{\psi_e}, \dfrac{\partial^2 F^{EMA}}{\partial (r_{t,\omega}^{b,e})^2} = -\dfrac{1}{a^{MT}} \\[2mm] \dfrac{\partial^2 F^{EMA}}{\partial (k_{t,\omega}^{s,h})^2} = -\dfrac{2}{\psi_h}, \dfrac{\partial^2 F^{EMA}}{\partial (r_{t,\omega}^{b,h})^2} = -\left(\eta^{HE}\right)^2 \left[\dfrac{1}{a^{GB}} + \dfrac{(\eta^a)^2}{a^{MT}}\right] \\[2mm] \dfrac{\partial^2 F^{EMA}}{\partial (k_{t,\omega}^{s,c})^2} = -\dfrac{2}{\psi_c}, \dfrac{\partial^2 F^{EMA}}{\partial (r_{t,\omega}^{b,c})^2} = -\left(\eta^{AR}\right)^2 \left[\dfrac{1}{a^{GB}} + \dfrac{(\eta^a)^2}{a^{MT}}\right] \end{cases} \tag{62}$$

where, $\eta^a = \eta^{WHB}\left(1 - \eta^{MT} - \eta^{loss}\right)\big/\eta^{MT}$, $\psi_e$, $\psi_h$, $\psi_c$, $a^{MT}$, $a^{GB}$, $\eta^{HE}$, $\eta^{AR}$ are all positive, so the second partial derivatives are all less than zero, and the function has a maximum point. When the constraints (37)–(39) are met, the EMA has a unique optimal solution. When the EMA can meet UAA energy demand without purchasing electricity from the power grid, the proof process is similar to the above, and this paper will not elaborate further. □

It should be mentioned that in this paper, we mainly consider the impact of wind and solar uncertainty and the IDR model on the master–slave game optimal scheduling of MAIES. For the robustness problem caused by parameter changes in the system, since the model of wind and solar uncertainty is considered in the optimal scheduling of MAIES, and some of the impacts of parameter changes can be eliminated by the proposed uncertainty model. At the same time, the IDR model is also taken into account in system operation. When the parameters change, the IDR model can effectively adjust the loads, so as to meet the energy consumption demand of users. Thus, the IDR model can also handle part of the influence caused by the changes in system parameters.

### 3.6. Solution Method

In order to solve the optimal scheduling problem of MAIES, a distributed equilibrium solution method of genetic algorithm combined with quadratic programming (GA-QP) is used in this paper. Because the decision-making model of leader EMA is a large-scale nonlinear optimization problem, using a genetic algorithm can reduce the complexity of the solution and improve the optimization ability [33]. Since the optimization objectives of followers EOA and UAA are quadratic functions, the quadratic programming method can be used to improve the speed and accuracy of the solution. Yalmip is a MATLAB toolbox for modeling and solving convex optimization problems. It provides a simple syntax that allows users to easily define optimization problems and solve them using a variety of built-in solvers. Yalmip supports a variety of solvers, such as CPLEX, Gurobi, MOSEK, etc. Gurobi is a powerful solver for solving linear problems, quadratic problems, mixed integer linear and quadratic problems, and supports multi-objective optimization. MOSEK is also recognized as one of the most efficient solvers for solving quadratic programming, second-order cone programming, and semidefinite programming. CPLEX is a highly optimized solver used for linear programming, integer programming, mixed integer programming, quadratic programming, constrained quadratic programming, second-order cone programming, and other large-scale complex optimization problems. In MATLAB, users can easily integrate with Cplex using the Yalmip interface. CPLEX has powerful solving performance and supports multiple optimization models, which is used in this paper to solve the optimal solution. When the quadratic programming is embedded into the iterative process of the genetic algorithm, the followers only need to accept the leader's price signal and feedback on their own power signal, which can effectively avoid the leakage of information and protect the information privacy and security of all agents [34]. Specifically, the genetic algorithm is used to initialize and update the selling and purchasing prices of the upper leader EMA, and the optimization problem of the lower followers is

solved by the CPLEX solver. The solving process of the GA-QP algorithm is shown in Figure 3.

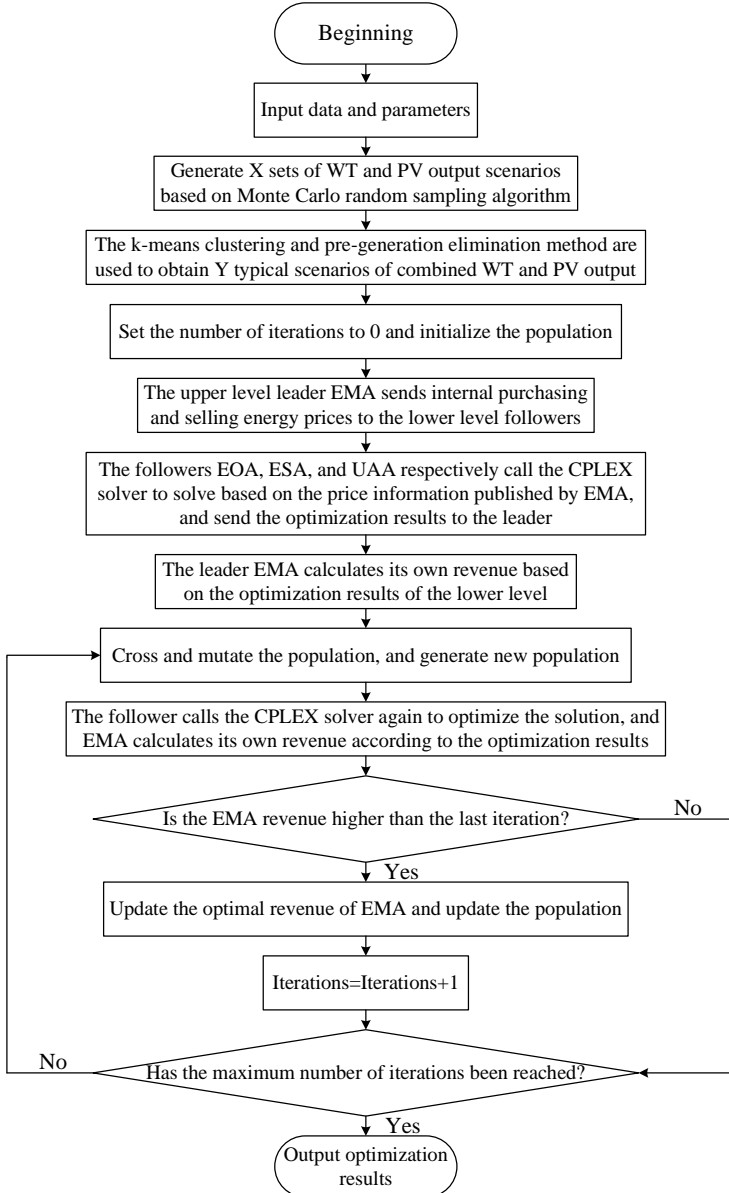

**Figure 3.** Model solving process.

## 4. Case Simulation

### 4.1. Parameter Setting

To verify the effectiveness of the method proposed in this paper, a certain combined cooling, heating, and power supply MAIES is used as an example. The simulation parameters are shown in Tables 1 and 2. The data of the electric, thermal, and cold loads come from a typical day in Northeast China. The data of the predicted electric, thermal, and cold loads are shown in Figure 4. The outdoor temperature and indoor fuzzy comfort temperature are shown in Figure 5. The time-sharing price of electricity purchased from the power grid is shown in Figure 6. The feed-in price of the power grid is 0.05 $/kWh. It should be mentioned that the feed-in price of the power grid and the time-sharing price of electricity purchased from the power grid are the lower and upper limits of electric price, so as to avoid followers directly trading with the power grid, resulting in the loss of the overall coordination role of the leader EMA. The upper and lower limits of heat price and cold price are 0.0714 $/kWh and 0.0286 $/kWh, respectively, which aim to ensure a normal

and reasonable market environment for energy interaction. The maximum average selling electric, heat, and cold prices of EMA are 0.1 \$/kWh, 0.0643 \$/kWh, and 0.0643 \$/kWh, respectively, so as to avoid EMA seeking the highest price for a long time. The constant coefficients of users' preference for electric, thermal, and cold energy $\zeta_e$, $\psi_e$, $\zeta_h$, $\psi_h$ and $\zeta_c$, $\psi_c$ are 0.0009, 1.5, 0.0011, 1.1 and 0.0012, 1.15, respectively, which can reflect users' demand preference for energy. It is assumed that the distance between the agents is small, and the influence of energy transmission line loss is not taken into account. Meanwhile, it is assumed that during the operation of the system, the input data and the proposed constraints and parameters are fully satisfied, and the special cases such as equipment failures are not considered.

The simulation calculations are based on MatlabR2018a software, using the YALMIP plug-in and calling the CPLEX solver to solve the optimization problem. The computer is configured with an Intel Core i7-10510U processor, main frequency 1.8 GHz, memory 8 GB. The population size is 30, the maximum number of iterations is 120 and the crossover probability is 0.9.

**Table 1.** Parameters of MAIES model.

| Parameter | Value | Parameter | Value | Parameter | Value |
|-----------|-------|-----------|-------|-----------|-------|
| $\gamma_p$ | 424 g/kWh | $a^{MT}$ | 0.0013 | $\eta^{loss}$ | 0.09 |
| $\gamma_h$ | 0.102 t/GJ | $b^{MT}$ | 0.16 | $\eta^{WHB}$ | 0.85 |
| $\delta^{e,h}$ | 6 MJ/kWh | $c^{MT}$ | 0 | $\eta^{HE}$ | 0.8 |
| $\sigma_e$ | 968 g/kWh | $a^{GB}$ | 0.0005 | $\eta^{AR}$ | 0.75 |
| $\sigma_g$ | 220 g/m$^3$ | $b^{GB}$ | 0.11 | $\eta^{GB}$ | 0.9 |
| $\lambda_{gas}$ | 9.78 kWh/m$^3$ | $c^{GB}$ | 0 | $\eta^{x,ch}$ | 0.9 |
| $\varepsilon_c$ | 40 \$/t | $\eta^{MT}$ | 0.41 | $\eta^{x,dis}$ | 0.85 |

**Table 2.** Parameters of MAIES constraints.

| Parameter | Value | Parameter | Value | Parameter | Value |
|-----------|-------|-----------|-------|-----------|-------|
| $E^{ES,\max}$ | 1600 kWh | $E^{ES,\min}$ | 400 kWh | $E^{HS,\max}$ | 1500 kWh |
| $E^{HS,\min}$ | 200 kWh | $P^{\max,ES,ch}$ | 350 kW | $P^{\max,ES,dis}$ | 350 kW |
| $P^{\max,HS,ch}$ | 350 kW | $P^{\max,HS,dis}$ | 300 kW | $P^{MT,up}$ | 400 kW |
| $P^{MT,down}$ | −400 kW | $P^{GB,up}$ | 500 kW | $P^{GB,down}$ | −400 kW |
| $P^{MT,\max}$ | 1200 kW | $H^{GB,\max}$ | 1000 kW | $P^{MT,\min}$ / $P^{GB,\min}$ | 0 kW |

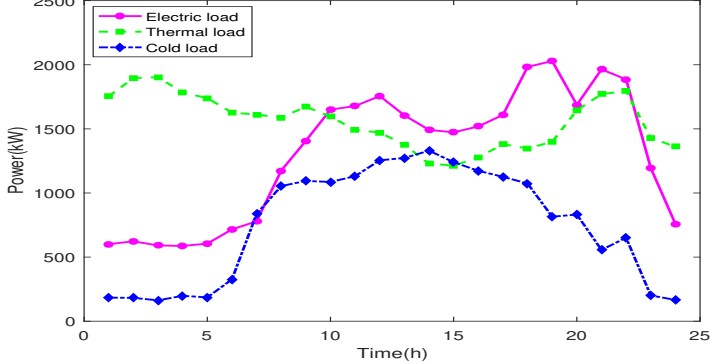

**Figure 4.** Predicted electric load, thermal load, and cold load.

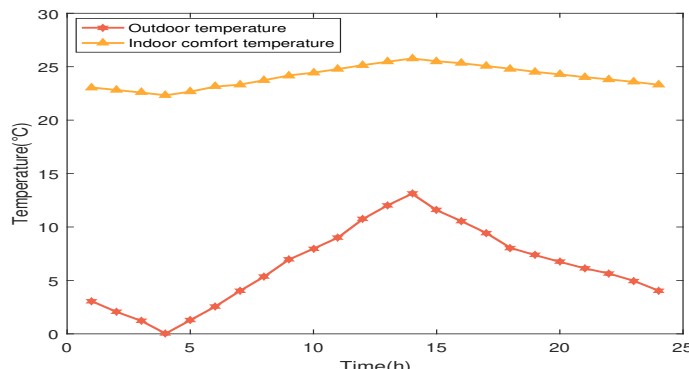

**Figure 5.** Indoor and outdoor temperature curves.

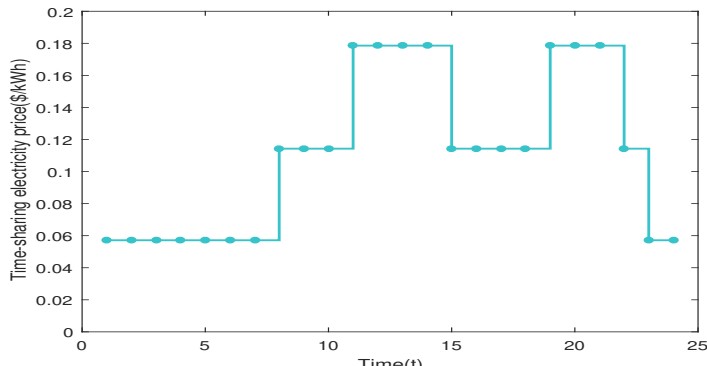

**Figure 6.** Time-sharing electricity price.

### 4.2. Analysis of Wind and Solar Uncertainty

According to the day-ahead prediction information of WT and PV output and the method proposed in this paper, the MC random sampling method is used to generate 1000 scenarios of WT and PV output, which are shown in Figure 7. Then, 10 WT and PV output scenarios with different probabilities are obtained by using the k-means clustering method. The Cartesian connection between WT and PV output scenarios is made, and the pre-generation elimination method is used to further reduce the scenarios. Finally, five typical scenarios of combined WT and PV output can be obtained. The scenarios of combined WT and PV output are shown in Figure 8, and the probability of each scenario is shown in Figure 9. It can be seen from Figure 8 that the generated scenarios of WT and PV output are all within the upper and lower boundaries. While ensuring a certain degree of difference, the generated scenarios can basically cover the fluctuation range of actual output. In addition, it can also be seen that the change trend of the WT output curves of the five scenarios is similar, and the change trend of the PV output curves is also similar. Therefore, the generation results of the scenarios reflect the uncertainty of WT and PV output, which can effectively simulate the characteristics of WT and PV output in MAIES and is beneficial for the overall planning and operation of the system.

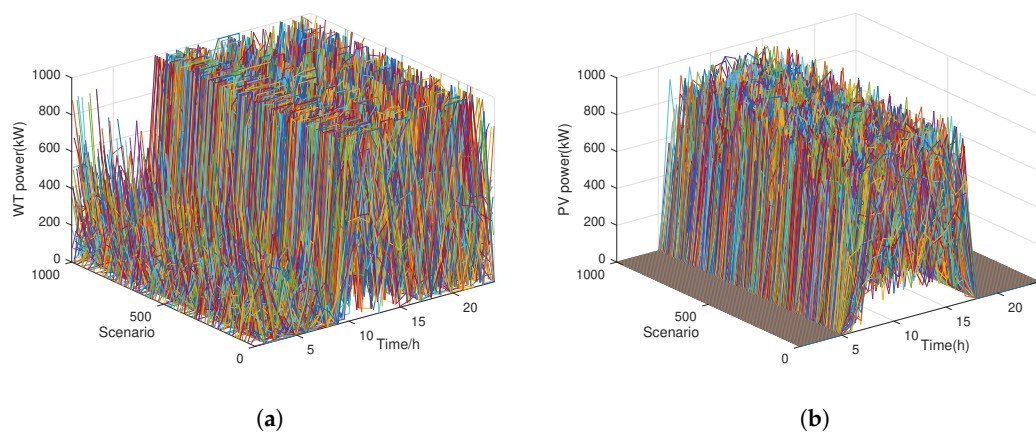

**Figure 7.** WT and PV output scenarios generated by MC: (**a**) WT output in 1000 scenarios; (**b**) PV output in 1000 scenarios.

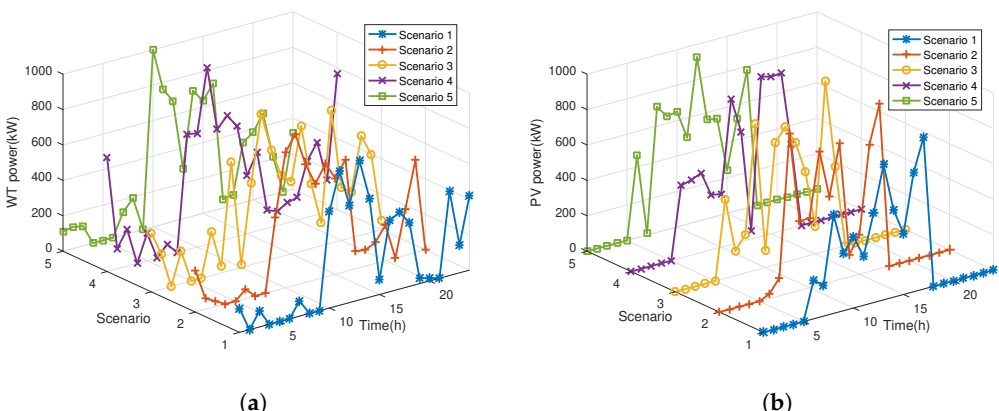

**Figure 8.** Combined WT and PV output scenarios after reduction: (**a**) WT output in 5 scenarios; (**b**) PV output in 5 scenarios.

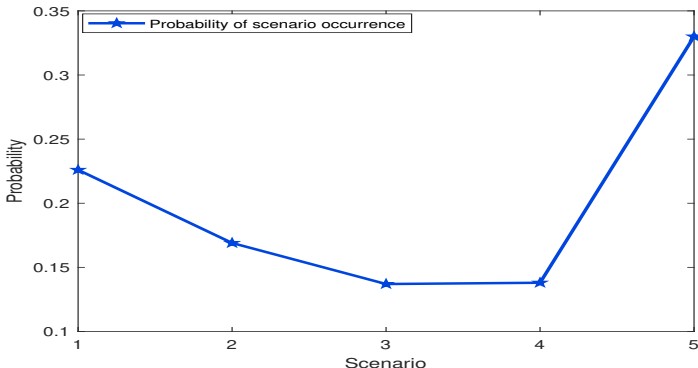

**Figure 9.** The probability of each scenario.

In order to analyze the influence of the uncertainty of WT and PV output on the system, scenario 6 and scenario 7 are added to the original five scenarios. Scenario 6 does not consider the uncertainty of the WT and PV output, that is, the day-ahead prediction data of WT and PV output is substituted into the model to calculate the revenue. Scenario 7 refers to the total benefit calculated by substituting the data of five scenarios into the model and multiplying the corresponding probability, which is the method used in this paper. The iterative curves of the revenues of each agent in scenario 7 are shown in Figure 10. The convergence process reflects the game process among the stakeholders. As the leader of the game, the EMA continuously adjusts its own purchasing and selling energy prices to

maximize its own revenue, and its revenue curve shows a gradually increasing trend. The followers EOA, ESA, and UAA also adjust their own strategies to achieve game equilibrium based on the price information released by the leader, and their revenue curves show a downward trend. When the number of iterations reaches about 60, the game among the agents reaches equilibrium. At this time, the revenues of EMA, EOA, ESA, and UAA are 1488.92\$, 684.38\$, 92.16\$, and 2831.85\$, respectively.

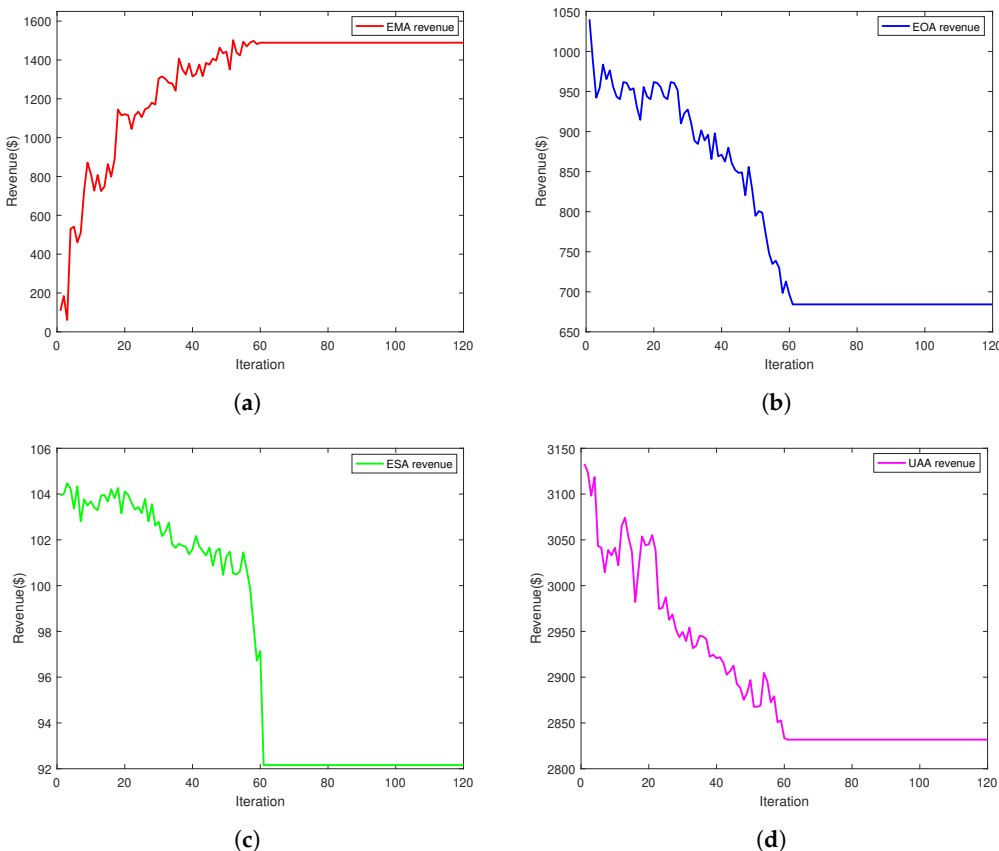

**Figure 10.** Iteration results of each agent: (**a**) Iteration curve of EMA revenue; (**b**) Iteration curve of EOA revenue; (**c**) Iteration curve of ESA revenu; (**d**) Iteration curve of UAA revenue.

Table 3 shows the comparative results of the revenue of each agent, the curtailment rate of wind power and photovoltaic power, and the calculation time in seven scenarios. It can be seen that the calculation time distribution of the seven scenarios is within the range of 10–12 min, which can meet the application needs of the actual scene. The proposed solution method effectively solves the optimization problem in each scenario. It can be seen that the difference in UAA revenue is relatively small in seven scenarios, which indicates that the uncertainty of WT and PV output has a relatively small impact on UAA decision-making. Compared with scenario 6 and scenario 7, the uncertainty of WT and PV output affect the power supply of EOA, and the revenues of EMA, EOA, and ESA increase by 49.35\$, 44.93\$, and 2.81\$, respectively, after considering the uncertainty of WT and PV output in scenario 7. There is a significant difference in the rates of wind and photovoltaic power curtailment between scenario 7 and scenario 6. The rates of wind and photovoltaic power curtailment in scenario 7 are 3.22% and 2.81% lower than those in scenario 6, respectively. Compared to the scenario 1 to 5, the rate of renewable energy curtailment in scenario 7 decreases by 3.34%, 0.47%, 1.25%, 0.83%, and 2.07%, respectively. Therefore, after considering the uncertainty of WT and PV output, the revenue of each agent has increased and the utilization rate of renewable energy has also been effectively improved, which plays a positive role in building the economically efficient and environmentally friendly energy system. In addition, the proposed uncertainty model can reduce the adverse effects

caused by the randomness of renewable energy output and improve the reliability and sustainability of the system operation.

**Table 3.** Comparison results of each scenario.

| Scenario | EMA Revenue ($) | EOA Revenue ($) | ESA Revenue ($) | UAA Revenue ($) | WT Power Curtailment (%) | PV Power Curtailment (%) | Calculation Time (s) |
|---|---|---|---|---|---|---|---|
| 1 | 1513.32 | 662.34 | 90.66 | 2820.33 | 6.12 | 4.26 | 647.64 |
| 2 | 1527.43 | 689.13 | 93.38 | 2830.06 | 4.85 | 2.66 | 622.56 |
| 3 | 1451.12 | 663.20 | 87.01 | 2834.52 | 5.21 | 3.08 | 608.50 |
| 4 | 1540.20 | 693.57 | 94.62 | 2818.65 | 4.98 | 2.89 | 673.08 |
| 5 | 1448.83 | 654.06 | 88.49 | 2825.50 | 5.77 | 3.34 | 621.33 |
| 6 | 1439.57 | 639.45 | 89.35 | 2814.32 | 7.85 | 5.22 | 659.74 |
| 7 | 1488.92 | 684.38 | 92.16 | 2831.85 | 4.63 | 2.41 | 641.12 |

### 4.3. Energy Price Optimization Results

The pricing strategy of EMA after game equilibrium is shown in Figure 11. In Figure 11a, the energy purchase and sale prices of EMA are always between the time-sharing price and the feed-in price of the external power grid, which aims to provide better prices for the energy supply side and energy consumption side. The trend of the selling price of EMA is close to the trend of time-sharing electricity price in the power grid, while the trend of the purchasing price of EMA is close to the changing trend of users' energy load, which aims to improve the enthusiasm of users to purchase electricity and the energy supply efficiency of EOA and reduce the electricity purchase from the power grid. Similarly, it can be seen from Figure 11b,c that the price trends of thermal and cold energy purchased by EMA are roughly the same as the actual thermal and cold load trends of users.

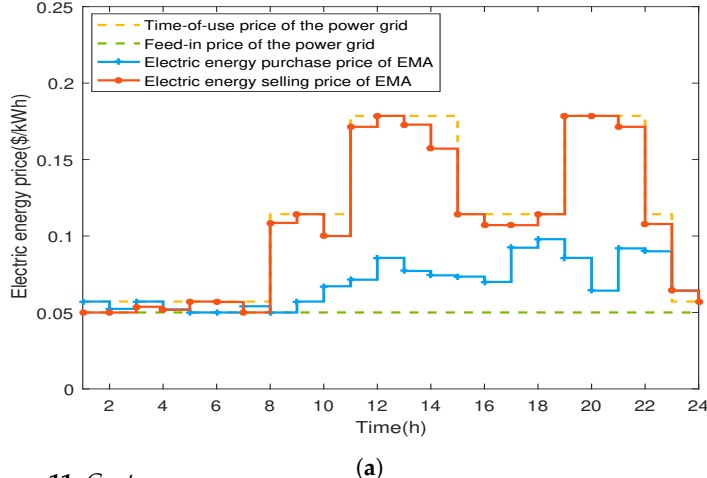

**(a)**

**Figure 11.** *Cont.*

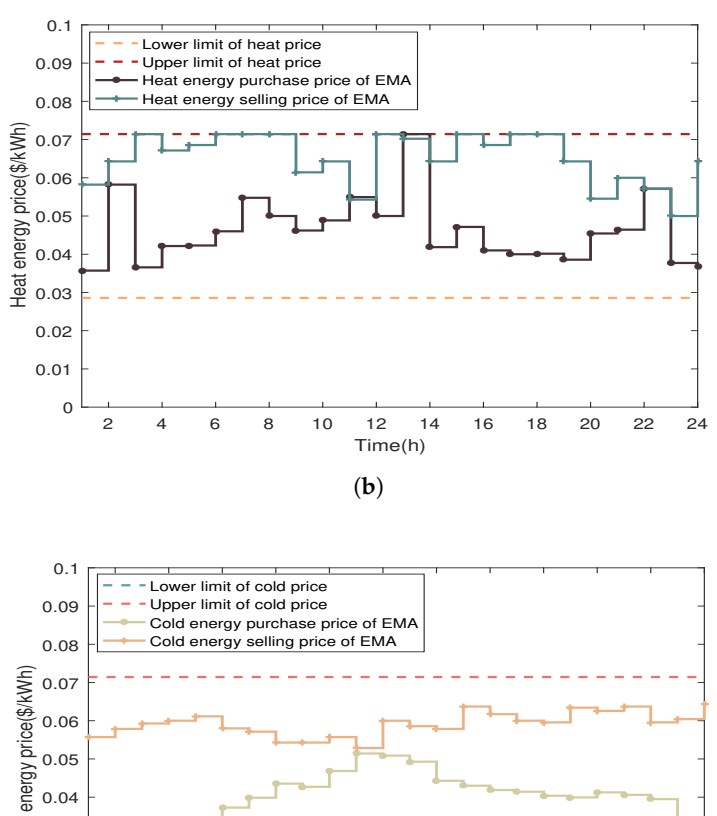

**Figure 11.** Optimization results of energy prices: (**a**) Electricity price curves; (**b**) Heat price curves; (**c**) Cold price curves.

*4.4. Comparative Analysis of Scenarios*

To further analyze the effect of the master–slave game optimal operation strategy of MAIES proposed in this paper considering the uncertainty of WT and PV output and the IDR on the benefits and carbon emissions of each agent, the following five comparative scenarios are set:

Scenario 1: the uncertainty of WT and PV output and the IDR are considered, the EMA, EOA, ESA, and UAA are considered, and the carbon trading model is taken into account in the optimal operation (the method proposed in this paper).

Scenario 2: the uncertainty of WT and PV output and the IDR are considered, and the EMA, EOA, ESA, and UAA are considered, but the carbon trading model is not taken into account in the optimal operation.

Scenario 3: the uncertainty of WT and PV output and the IDR are not considered, and the EMA, EOA, ESA, and UAA are considered, and the carbon trading model is taken into account in the optimal operation.

Scenario 4: the uncertainty of WT and PV output and the IDR are considered, the EMA, EOA, and UAA are considered, and the carbon trading model is taken into account in the optimal operation.

Scenario 5: the uncertainty of WT and PV output and the IDR are not considered, the EMA, EOA, and UAA are considered, and the carbon trading model is not taken into account in the optimal operation.

The operation results of the five scenarios are shown in Tables 4 and 5. As can be seen from Table 4, the calculation time for all 5 scenarios is within the range of 7–11 min, which can meet the demands of practical applications. The optimal scheduling problem in each scenario is effectively solved by the proposed solution method. Comparing scenario 1 and scenario 2, it can be seen that the revenues of EMA, EOA and ESA of scenario 2 are 17.66$, 80.87$, and 2.27$ higher than those in scenario 1, respectively. The reason is that the carbon trading mechanism is not considered in scenario 2, and there is a lack of carbon trading cost constraints in the EMA, EOA, and ESA models. Furthermore, the carbon emissions of EOA in scenario 2 are 1975.08 kg higher than that in scenario 1, which is because carbon emission constraints are not taken into account, and the EOA will produce more energy and sell it to EMA, thus reducing energy purchase of EMA from the power grid and reducing carbon emissions generated by EMA. Therefore, when the carbon trading mechanism is not considered, although the revenues of EMA and EOA will be increased to some extent, the carbon emissions of the system will increase significantly. Comparing scenario 1 and scenario 3, according to the analysis in Section 4.2, the EOA revenue will increase by considering the uncertainty of WT and PV output. However, the IDR is not considered in scenario 3, and the thermal load cannot be reduced. So compared to scenario 1, the EMA needs to purchase more thermal power from EOA to meet the thermal demand of UAA in scenario 3. In this way, the thermal power output of EOA units increases and the carbon emissions and carbon trading costs also increase, and the EOA revenue increases by 1314.45$ compared with scenario 1. In addition, due to the IDR not being considered in scenario 3, the users cannot change the energy consumption behavior of electric and cold load during the peak periods of electric and cold prices, which also leads to an increase in the costs of electric and cold energy consumption. So the UAA revenue is 385.06$ less than that of scenario 1. Comparing scenario 1 and scenario 4, it can be seen that compared with scenario 1, the revenues of EMA, EOA, and UAA in scenario 4 decrease by 1417.85$, 157.91$, and 136.22$, respectively. This is because adding energy storage devices in scenario 1 may occupy a small share of user energy purchases in EMA, but the ESA can not only alleviate the output pressure of the devices in EOA by storing and releasing energy but also reduce the energy purchasing cost of direct interaction between EMA and the power grid during peak load periods. In addition, it can also provide users with more favorable energy purchase prices than EMA, and reduce the energy purchase costs of users. The carbon emissions of EMA and EOA in scenario 4 are 558.25 kg and 2405.85 kg more than those in scenario 1, respectively. The reason is that the lack of energy storage devices increases the output of the EOA devices and the EMA purchases more energy from the grid, which both result in the increase in carbon emissions. Comparing scenario 4 and scenario 5, it can be seen that the UAA revenue in scenario 5 is 372.76$ less than that in scenario 4. The reason is that the IDR is not considered in scenario 5, which leads to users being unable to obtain subsidies and flexibly adjust energy demand to use lower energy prices. It also can be seen that the carbon emissions of EMA and EOA in scenario 5 are 278.55 kg and 644.27 kg more than those in scenario 4, which is due to the fact that the lack of carbon emissions constraints in scenario 5 increases the carbon emissions of the system.

Therefore, by considering IDR in the optimal scheduling of MAIES, users actively participate in the adjustment of energy load, which enables users to gain more benefits and improve their energy usage satisfaction. The IDR model can realize peak shaving and valley filling of loads, which can effectively reduce the energy supply pressure of the power grid and ensure the safe and stable operation of the power grid. Furthermore, by taking into account the carbon trading model, the carbon emissions generated by system operation can be effectively limited, and have an important effect on environmental protection and sustainable development. In today's society that vigorously advocates and promotes carbon emission reduction, the carbon trading market will continue to be a hot topic and a long-term energy policy. So this paper makes a detailed study on the effect of the carbon trading model in MAIES operation and provides some research basis and theoretical support for the future development of carbon trading mechanism.

**Table 4.** Revenue of each agent in different scenarios.

| Scenario | EMA Revenue ($) | EOA Revenue ($) | ESA Revenue ($) | UAA Revenue ($) | Calculation Time (s) |
|---|---|---|---|---|---|
| 1 | 1488.92 | 684.38 | 92.16 | 2831.85 | 641.12 |
| 2 | 1506.58 | 765.25 | 94.43 | 2826.36 | 583.77 |
| 3 | 1548.33 | 887.19 | 89.52 | 2446.79 | 516.06 |
| 4 | 1471.07 | 526.47 | — | 2695.63 | 556.41 |
| 5 | 1497.92 | 602.17 | — | 2322.87 | 469.38 |

**Table 5.** Carbon emissions and costs in different scenarios.

| Scenario | EMA Emissions (kg) | EOA Emissions (kg) | EMA Carbon Cost ($) | EOA Carbon Cost ($) |
|---|---|---|---|---|
| 1 | 1304.57 | 9917.41 | 20.10 | 152.84 |
| 2 | 1152.01 | 11,892.49 | — | — |
| 3 | 1325.34 | 11,231.86 | 20.42 | 173.09 |
| 4 | 1862.82 | 12,323.26 | 28.71 | 189.91 |
| 5 | 2141.37 | 12,967.53 | — | — |

*4.5. Load Optimization Results*

The electric load, thermal load, and cold load before and after considering IDR are shown in Figure 12. As can be seen from Figure 12a, under the effect of electricity price, in order to reduce the total electric cost, the electric load curve exhibits the characteristic of peak shaving and valley filling before and after IDR. The two peak periods of the original electric load appear at 11:00–12:00 and 18:00–22:00, and the electricity price is high in these periods. After the optimization of the IDR on the user side, the peak load significantly decreases and shifts to the load valley stage, in which the electricity price is low at 0:00–8:00 and 23:00–24:00, and the load fluctuation decreases obviously. As can be seen from Figure 12b, after considering the thermal demand response based on incentive in the system operation, the thermal load was reduced and shifted and the thermal load curve has become smoother, so as to enable users to gain more benefits. From Figure 12c, it can be seen that when considering the cold demand response model based on fuzzy comfort, on the premise of satisfying the living comfort of users, the users actively change the behavior of cold energy consumption, thus reducing energy consumption and increasing revenue. Considering Tables 4 and 5, it can be seen that the IDR strategy proposed in this paper can effectively improve the comprehensive benefits of multiple subjects and achieve a win-win situation in the entire energy market.

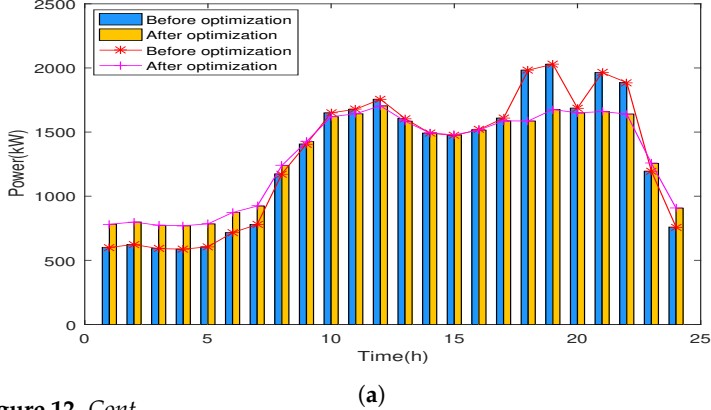

(a)

**Figure 12.** *Cont.*

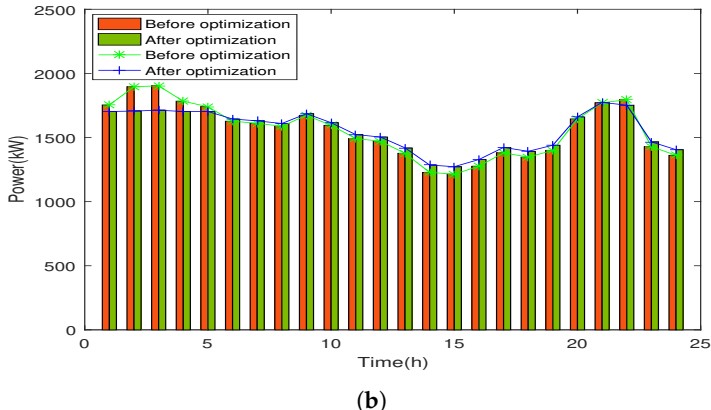

(**b**)

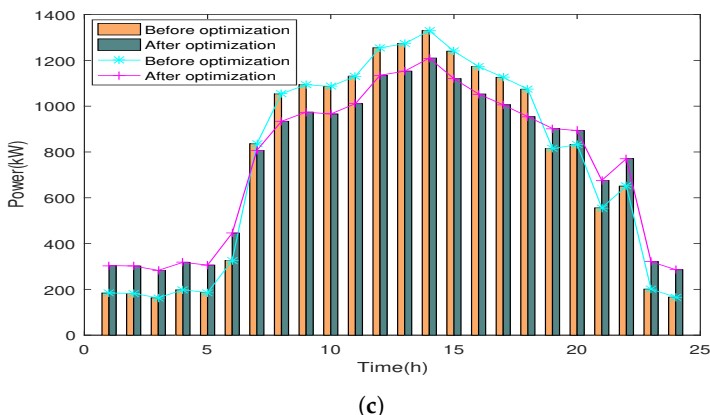

(**c**)

**Figure 12.** Variations of load before and after considering IDR: (**a**) Variations of electric load; (**b**) Variations of thermal load; (**c**) Variations of cold load.

### 4.6. Supply and Demand Balance Results

The results of electric, thermal, and cold energy balance after Stackelberg game optimization are shown in Figure 13. At 23:00–07:00, the electric load and cold load are relatively few, and the electricity price is in the valley range. The electric load is mainly met by WT, and the insufficient part is supplemented by MT and electricity purchased from the power grid. The EOA increases the output of MT to make more profits, and the excess electric energy is stored by the ES of ESA. Furthermore, the GB and WHB work together to satisfy the thermal energy demand of UAA. If the thermal demand cannot be met, the HS of ESA will supplement it by releasing and storing thermal energy. The cold load demand is relatively low, which is mainly provided by ISAC. At 08:00–10:00 and 14:00–17:00, as the electric load and cold load gradually increase, the PV and WT output of EOA are completely accommodated, and the MT output increases. However, the EOA restricts the output of MT and GB by comparing energy sales revenue and operating cost, so part of the load demand cannot be met by EOA. Therefore, the energy demand of UAA also be met by purchasing more electricity from the grid by EMA. The cold load is satisfied by ISAC with lower power consumption through air conditioning mode and ice-melting refrigeration at the same time, and the insufficient part is supplemented by AR. The thermal load is still satisfied by GB and WHB, and the insufficient part is met by the HS. At 11:00–13:00 and 18:00–23:00, the electric load is in the peak periods, and the electric load with insufficient energy supply is met by ES discharge and purchasing more electricity from the power grid. The refrigeration capacity of AR increases, and together with ISAC meets the cold load demand.

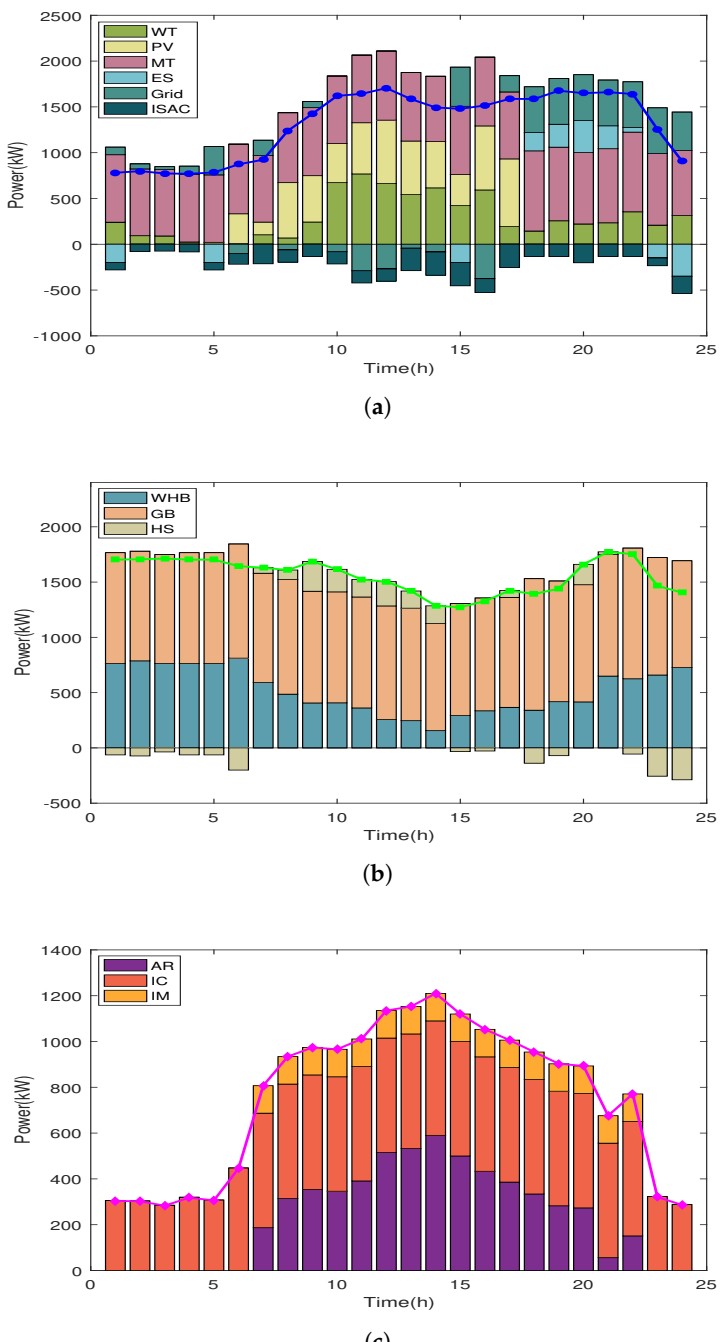

**Figure 13.** Supply and demand balance results: (**a**) Electric power balance; (**b**) Thermal power balance; (**c**) Cold power balance.

## 5. Conclusions and Future Works

This paper proposes a master–slave game optimal scheduling strategy of MAIES considering the wind and solar uncertainty and the IDR. The conclusions are as follows:

(1) After considering the wind and solar uncertainty model, the benefits of EMA, EOA, ESA, and UAA increased by 49.35\$, 44.93\$, 2.81\$ and 17.53\$, respectively. The rates of WT power curtailment and PV power curtailment were reduced by 3.22% and 2.81%, respectively. So the wind and solar uncertainty model proposed in this paper is beneficial for accommodating renewable energy and improving the operational economy of MAIES.

(2) The multi-load and multi-type IDR model can realize peak shaving and valley filling of electric load, thermal load, and cold load within a reasonable range. After

considering the IDR model, the revenue of the users increased by 385.06$. At the same time, by taking into account the carbon trading mechanism, the carbon emissions generated by system operation were reduced by 1822.52 kg.

(3) The master–slave game optimization model of MAIES guides the controllable device output of EOA, the energy storage and release power of ESA, and the energy consumption strategy of UAA through the reasonable price information released by EMA, and realizes the cooperative optimal scheduling of multiple agents and multiple energy sources.

In future works, the robust optimal scheduling of MAIES based on game theory will be studied, so as to solve the robustness problem caused by parameter changes in the system. In addition, we plan to study the dynamic nature of MAIES, which includes discussing how the system evolves, transitions between different operational states, and responds to external factors. The variations of energy supply and demand at different time intervals in MAIES must also be considered in the next stage of our work.

**Author Contributions:** Conceptualization, B.Z. and D.W.; methodology, B.Z.; software, B.Z. and D.W.; validation, B.Z. and D.W.; formal analysis, B.Z.; writing—original draft preparation, B.Z. and D.W.; writing—review and editing, B.Z. and D.W. All authors have read and agreed to the published version of the manuscript.

**Funding:** This research received no external funding.

**Institutional Review Board Statement:** Not applicable.

**Data Availability Statement:** Data are contained within the article.

**Conflicts of Interest:** The authors declare no conflicts of interest.

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
