# Peer review of "Master–Slave Game Optimal Scheduling for Multi-Agent Integrated Energy System Based on Uncertainty and Demand Response"

_sustainability, doi:10.3390/su16083182_

Round 1

Reviewer 1 Report

Comments and Suggestions for Authors

The manuscript presents a comprehensive study on optimizing Microgrid-based Autonomous Integrated Energy Systems (MAIES) considering factors like wind and solar uncertainty, Interruptible Demand Response (IDR), and carbon trading mechanisms. The proposed master-slave game optimization model shows promise in enhancing system efficiency and economic benefits. However, there are areas where the manuscript could be strengthened. Therefore, the authors should address the following comments:

  1. In the abstract, the authors briefly mentioned case simulations that demonstrated the effectiveness of the proposed methods, but they lacked specific results or insights gained from these simulations. 
  2. The authors should ensure consistency in terminology throughout the introduction and the rest of the manuscript. For example, "integrated energy system" and "multi-energy system" are used interchangeably, which could cause confusion.
  3. While the authors in the introduction section identified the gap in existing research related to conflicts of interest and demand response behavior in multi-energy systems, further elaboration on why addressing this gap is essential would strengthen the rationale for the proposed research.
  4. The authors should provide a more explicit justification for using game theory to model the interaction among stakeholders in MAIES. Addressing this point would strengthen the rationale for the proposed approach.
  5. In the introduction section, the authors effectively reviewed the related research on optimization strategies for IES. However, further engagement with the strengths and limitations of these approaches could provide a more nuanced understanding of the research landscape and highlight the unique contributions of the proposed approach.
  6. While the section on the Master-slave Game Framework describes the roles of individual agents, it would be beneficial to elaborate on the interactions among these agents within the master-slave game framework. 
  7. While the authors provide extensive detail on the simulation parameters and constraints, it would be beneficial to briefly discuss any assumptions made or external factors considered in setting these parameters.
  8. Ensuring that the chosen parameters and constraints align with the research objectives is important. The authors should briefly discuss how the selected parameters contribute to testing the effectiveness of the proposed method in optimizing the MAIES.
  9. The authors should consider discussing any sensitivity analysis conducted on the parameters to assess the robustness of the simulation results.
  10. The key observations regarding revenue generation and carbon emissions across scenarios are valuable. However, the section could delve deeper into the implications of these findings for stakeholders, particularly in terms of long-term sustainability, market competitiveness, and policy implications. 
  11. The section on the Impact of Thermal Demand Response convincingly illustrates the impact of thermal demand response on reducing and smoothing thermal loads. However, further elaboration on the specific mechanisms through which thermal demand response achieves these outcomes, such as load shifting or load shedding strategies, would provide readers with deeper insights into the operational dynamics of the system.
  12. While the conclusions mention increased benefits for each agent, it would be beneficial to provide specific quantitative metrics or performance indicators to quantify these benefits. This could include measures such as cost savings, revenue increases, or reductions in carbon emissions achieved through the proposed optimization strategy.
Comments on the Quality of English Language

The quality of the English language in the manuscript is generally good. The text is coherent and the ideas are clearly expressed. However, there are some instances of awkward phrasing and grammatical errors that could be improved for better readability and clarity. 

Author Response

Responses to Reviewer

Dear reviewer:

The authors would like to express their gratitude for your timely and competent reviews. We have carefully read the comments, and revised our paper according to the comments and suggestions.

Response to Comments

Point 1: In the abstract, the authors briefly mentioned case simulations that demonstrated the effectiveness of the proposed methods, but they lacked specific results or insights gained from these simulations.

Response 1: The comment is greatly appreciated. Thank you for your kind suggestions. The specific results obtained from the case simulations have been added to the abstract, which are marked in red.

Point 2: The authors should ensure consistency in terminology throughout the introduction and the rest of the manuscript. For example, "integrated energy system" and "multi-energy system" are used interchangeably, which could cause confusion.

Response 2: The comment is greatly appreciated. Thank you for your kind suggestions. The “multi-energy system” has been uniformly rewritten as “integrated energy system(IES)”. And we have carefully proofread the full text and unified the terminology in the paper.

Point 3: While the authors in the introduction section identified the gap in existing research related to conflicts of interest and demand response behavior in multi-energy systems, further elaboration on why addressing this gap is essential would strengthen the rationale for the proposed research.

Response 3: The comment is greatly appreciated. Thank you for your kind suggestions. We have further explained and supplemented the reasons for addressing the gap, which are marked in red in introduction section.

Point 4: The authors should provide a more explicit justification for using game theory to model the interaction among stakeholders in MAIES. Addressing this point would strengthen the rationale for the proposed approach.

Response 4: The comment is greatly appreciated. Thank you for your kind suggestions. We have further explained and supplemented the justifications for using game theory to model the interaction among stakeholders in the third paragraph of introduction section, which are marked in red. And we have also further supplemented the theory of master-slave game in the first half of Sub-section 3.5, which are marked in red.

Point 5: In the introduction section, the authors effectively reviewed the related research on optimization strategies for IES. However, further engagement with the strengths and limitations of these approaches could provide a more nuanced understanding of the research landscape and highlight the unique contributions of the proposed approach.

Response 5: The comment is greatly appreciated. Thank you for your kind suggestions. The strengths and limitations of the approaches of related research on optimization strategies for IES have been further explained and discussed in introduction section, which are marked in red. 

Point 6: While the section on the Master-slave Game Framework describes the roles of individual agents, it would be beneficial to elaborate on the interactions among these agents within the master-slave game framework.

Response 6: The comment is greatly appreciated. Thank you for your kind suggestions. We have further supplemented and explained the interaction relationship among the agents within the master-slave game framework proposed in this paper, which are marked in red in Sub-section 2.1.

Point 7: While the authors provide extensive detail on the simulation parameters and constraints, it would be beneficial to briefly discuss any assumptions made or external factors considered in setting these parameters.

Response 7: The comment is greatly appreciated. Thank you for your kind suggestions. We have added the assumptions made in the simulation part, which are marked in red in Sub-section 4.1. 

Point 8: Ensuring that the chosen parameters and constraints align with the research objectives is important. The authors should briefly discuss how the selected parameters contribute to testing the effectiveness of the proposed method in optimizing the MAIES.

Response 8: The comment is greatly appreciated. Thank you for your kind suggestions. We have added the discussions about how the selected parameters contribute to testing the effectiveness of the proposed method, which are marked in red in Sub-section 4.1. 

Point 9: The authors should consider discussing any sensitivity analysis conducted on the parameters to assess the robustness of the simulation results.

Response 9: The comment is greatly appreciated. Thank you for your kind suggestions. In this paper, we mainly consider the impact of wind and solar uncertainty and the integrated demand response mechanism on the master-slave game optimal scheduling of multi-agent integrated energy system. And in the simulation section, we also set up comparative scenarios in different situations and analyze simulation results from multiple perspectives, so as to verify the effectiveness of the methods proposed in this paper. For the robustness problem caused by parameter changes in the system, on the one hand, the model of wind and solar uncertainty is considered in the optimal scheduling of multi-agent integrated energy system, which aims to reduce the adverse effects of randomness of distributed energy generation. So when the parameters change in the system, the proposed uncertainty model can eliminate some of the impact of parameter changes on system operation.

On the other hand, the integrated demand response mechanism is considered in the optimal scheduling of multi-agent integrated energy system. According to different scheduling potential and flexible characteristics of loads, the electric load demand response model based electricity price, the thermal load demand response model based economic incentive and the cold load demand response model based on fuzzy comfort are established respectively. When the parameters in the system change, the integrated demand response model can effectively adjust the loads, so as to meet the energy consumption demand of users. Thus the integrated demand response model in this paper can also deal with part of the effect caused by the changes of system parameters.

We have added some statements and supplements at the end of Sub-section 3.5 to explain these points more clearly, which are marked in red.

Your suggestions are very enlightening to us and provide a direction for the next stage of our work. The robust optimal scheduling of multi-agent integrated energy system based on game theory is the focus of our research in the next stage, so as to solve the robustness problem caused by parameter changes in the system. And the prospects of future works have been added to Section 5, which is marked in red. 

Point 10: The key observations regarding revenue generation and carbon emissions across scenarios are valuable. However, the section could delve deeper into the implications of these findings for stakeholders, particularly in terms of long-term sustainability, market competitiveness, and policy implications.  

Response 10: The comment is greatly appreciated. Thank you for your kind suggestions. The implications of the findings are further explained and discussed in Sub-section 4.2 and Sub-section 4.4, which are marked in red.

Point 11: The section on the Impact of Thermal Demand Response convincingly illustrates the impact of thermal demand response on reducing and smoothing thermal loads. However, further elaboration on the specific mechanisms through which thermal demand response achieves these outcomes, such as load shifting or load shedding strategies, would provide readers with deeper insights into the operational dynamics of the system.

Response 11: The comment is greatly appreciated. Thank you for your kind suggestions. We have further explained and supplemented the specific mechanisms for achieving load transfer and load reduction through thermal load demand response, which are marked in red in Sub-section 2.5.2.

Point 12: While the conclusions mention increased benefits for each agent, it would be beneficial to provide specific quantitative metrics or performance indicators to quantify these benefits. This could include measures such as cost savings, revenue increases, or reductions in carbon emissions achieved through the proposed optimization strategy.

Response 12: The comment is greatly appreciated. Thank you for your kind suggestions. We have added some specific quantitative metrics and performance indicators to quantify the benefits of the methods proposed in this paper, which are marked in red in Section 5.

Point 13: The quality of the English language in the manuscript is generally good. The text is coherent and the ideas are clearly expressed. However, there are some instances of awkward phrasing and grammatical errors that could be improved for better readability and clarity.  

Response 13: The comment is greatly appreciated. Thank you for your kind suggestions. We have carefully checked the full text and corrected the phrasing and grammatical errors in the paper.

Reviewer 2 Report

Comments and Suggestions for Authors

The manuscript is well structured and the basic information from the available literature is efficiently summarized in the introduction. Nevertheles, as the manuscript is a litle bit too long, chapter 2 can be more concicly presented.

Review report

Manuscript ID: sustainability-2920442

Title: Master-slave Game Optimal Scheduling for Multi-agent Integrated Energy System Based on Uncertainty and Demand Response

The aim of this manuscript is to proposes an optimal scheduling strategy of multi-agent integrated energy system (MAIES) considering the wind and solar uncertainty and the IDR. The authors proposed and tested two-level game model (one master and multiple slaves) with the goal of maximizing the revenue of each agent. The Weibull and Beta distribution is used to simulate wind power generation and respectively the photovoltaic power generation. Monte Carlo method is applied to generate random scenarios which, to promote the utilization of renewable energy and improve the economy of system operation, are reduced by applying k-means clustering method and pre-generation elimination technology.

The authors concisely resume the literature data on integrated energy system (IES), and presented the principes of the master-slave game in the frame of MAIES and the mathematical models used.

As the manuscript is a litle bit too long, chapters 2.2 to 2.5 could be more concisely presented, and the necessary details that can be found in the literature can be cited and not detailed presented in these chapters.

The conclusions are adequately supported by the modeling results on the combined cooling, heating and power supply MAIES that was used as an example to verify the effectiveness of the method proposed by the authors.

English is mostly adequate, but some checks are needed as there are some mistakes in using the indefinite article a/an (i.e. on line 28 – an significant, on line 29 a optimal). In some sentences the predicate is missing i.e. on lines 33-34 „And through the complementarity of various energy sources to solve the problem of distribution network congestion.

Comments on the Quality of English Language

The manuscript is written in an English that is easy to understand. Small errors related to the use of the indefinite article a/an. For example you used an significant (line 28), an optimal (line 29). 

Author Response

Responses to Reviewer

Dear reviewer:

The authors would like to express their gratitude for your timely and competent reviews. We have carefully read the comments, and revised our paper according to the comments and suggestions.

Response to Comments

Point 1: The manuscript is well structured and the basic information from the available literature is efficiently summarized in the introduction. Nevertheles, as the manuscript is a litle bit too long, chapter 2 can be more concicly presented.

Response 1: The comment is greatly appreciated. Thank you for your kind suggestions. Thank you for your approval of our paper. We have made modifications to chapter 2 to make it more concise and clear. And we have also carefully examined the full text and further refined the language and expression of this paper.

Point 2: As the manuscript is a litle bit too long, chapters 2.2 to 2.5 could be more concisely presented, and the necessary details that can be found in the literature can be cited and not detailed presented in these chapters.

Response 2: The comment is greatly appreciated. Thank you for your kind suggestions. We have made modifications to chapter 2 to make it more concise and clear.

Point 3: English is mostly adequate, but some checks are needed as there are some mistakes in using the indefinite article a/an (i.e. on line 28 – an significant, on line 29 a optimal). In some sentences the predicate is missing i.e. on lines 33-34 „And through the complementarity of various energy sources to solve the problem of distribution network congestion.”

Response 3: The comment is greatly appreciated. Thank you for your kind suggestions. These mistakes have been corrected. And we have carefully checked the full text and corrected the phrasing and grammatical errors in the paper.

Point 4: The manuscript is written in an English that is easy to understand. Small errors related to the use of the indefinite article a/an. For example you used an significant (line 28), an optimal (line 29).  

Response 4: The comment is greatly appreciated. Thank you for your kind suggestions. These mistakes have been corrected. And we have carefully checked the full text and corrected the phrasing and grammatical errors in the paper.

Reviewer 3 Report

Comments and Suggestions for Authors

The paper contributes to understanding decision-making processes in IES and offers a framework for enhancing system efficiency and sustainability. The model provides a comprehensive approach to energy management in MAIES by considering uncertainties, demand response, and human factors.

Advised amendments:

1.      Clarify Software Utilization: Elaborate on the software tools used for uncertainty modeling, optimization, and simulation in the MAIES framework, specifying the rationale behind the software selection and its advantages. Consider providing examples or case studies demonstrating the application of selected software in similar energy system optimization projects.

2.      Simulation Tools: Discuss using simulation tools, including software packages like GAMS, MATLAB, or Python-based libraries, to assess the MAIES model’s performance under different scenarios and parameter configurations in Section 4, Case Simulation.

3.      Enhance Discussion on Game Theory Applications: Expand the discussion on game theory applications, particularly Stackelberg games, in optimizing energy management strategies within integrated energy systems. Provide insights into how Stackelberg games effectively coordinate actions among multiple agents in MAIES and potential challenges and strategies for addressing them in real-world applications.ù

4.      Dynamic nature of IES: Emphasize the dynamic nature of IES in the paper. This includes discussing how the system evolves, transitions between different operational states, and responds to external factors such as renewable energy generation variability and demand fluctuations. The author is encouraged to provide insights into how the dynamics of IES are modeled in the analysis. This could involve discussing the representation of state variables, transition probabilities, and system dynamics equations used in simulation or optimization models. Temporal aspects, such as i. time-dependent constraints, ii. scheduling decisions, and iii. optimization objectives are certainly already considered in the paper, but the authors fail to account for energy supply and demand variations over different time intervals. By addressing these points, the author can provide a more comprehensive understanding of the dynamic behavior of IES and how it influences system operation and management strategies.

Minor Amendments:

·        Amend “generated flue gas” on line 134 to correct the misprint.

·        If carbon emissions reduction is a significant aspect of the research or mentioned in the source material, reference it explicitly. Otherwise, if relevant to the topic, consider including it in the context of environmental sustainability and energy system optimization.

Author Response

Responses to Reviewer

Dear reviewer:

The authors would like to express their gratitude for your timely and competent reviews. We have carefully read the comments, and revised our paper according to the comments and suggestions.

Response to Comments

Point 1: Clarify Software Utilization: Elaborate on the software tools used for uncertainty modeling, optimization, and simulation in the MAIES framework, specifying the rationale behind the software selection and its advantages. Consider providing examples or case studies demonstrating the application of selected software in similar energy system optimization projects.

Response 1: The comment is greatly appreciated. Thank you for your kind suggestions. We have further explained and supplemented the software utilization and software tools in this paper, which are marked in red in Sub-section 3.6 and Sub-section 4.1. 

Point 2: Simulation Tools: Discuss using simulation tools, including software packages like GAMS, MATLAB, or Python-based libraries, to assess the MAIES model’s performance under different scenarios and parameter configurations in Section 4, Case Simulation.

Response 2: The comment is greatly appreciated. Thank you for your kind suggestions. We have further discussed and explained the simulation tools used in this paper, which are marked in red in Sub-section 3.6 and Sub-section 4.1. And the calculation time of each scenario has also been added to Tables 3 and Tables 4. The relevant explanations and explanations have also been added to Sub-section 4.2 and Sub-section 4.4, which are marked in red.

Point 3: Enhance Discussion on Game Theory Applications: Expand the discussion on game theory applications, particularly Stackelberg games, in optimizing energy management strategies within integrated energy systems. Provide insights into how Stackelberg games effectively coordinate actions among multiple agents in MAIES and potential challenges and strategies for addressing them in real-world applications.

Response 3: The comment is greatly appreciated. Thank you for your kind suggestions. We have further enhanced the discussion on game theory applications, which are marked in red in the third paragraph of the Section 1, the first half of the Sub-section 3.5 and the Sub-section 2.1.  

Point 4: Dynamic nature of IES: Emphasize the dynamic nature of IES in the paper. This includes discussing how the system evolves, transitions between different operational states, and responds to external factors such as renewable energy generation variability and demand fluctuations. The author is encouraged to provide insights into how the dynamics of IES are modeled in the analysis. This could involve discussing the representation of state variables, transition probabilities, and system dynamics equations used in simulation or optimization models. Temporal aspects, such as i. time-dependent constraints, ii. scheduling decisions, and iii. optimization objectives are certainly already considered in the paper, but the authors fail to account for energy supply and demand variations over different time intervals. By addressing these points, the author can provide a more comprehensive understanding of the dynamic behavior of IES and how it influences system operation and management strategies.  

Response 4: The comment is greatly appreciated. Thank you for your kind suggestions. It is a really good idea to study the dynamic nature of IES and the variations of energy supply and demand at different time intervals. Your suggestions are very enlightening to us and provide a direction for the next stage of our work. We have added our future work plans mentioned above to the Section 5, which are marked in red in Section 5. 

Point 5: Amend “generated flue gas” on line 134 to correct the misprint.

Response 5: The comment is greatly appreciated. Thank you for your kind suggestions. This misprint has been corrected. And we have carefully checked the full text and corrected the phrasing and grammatical errors in the paper.

Point 6: If carbon emissions reduction is a significant aspect of the research or mentioned in the source material, reference it explicitly. Otherwise, if relevant to the topic, consider including it in the context of environmental sustainability and energy system optimization.

Response 6: The comment is greatly appreciated. Thank you for your kind suggestions. The carbon emissions reduction is a significant aspect of the research. And we have added the reference of the carbon emissions reduction, which are marked in red in Section 1 and Sub-section 2.3. 

Round 2

Reviewer 3 Report

Comments and Suggestions for Authors

Subject: Review Feedback on Manuscript ID: Sustainability_2920442 

Dear Authors, dear SUSTAINABILITY Editorial Office, 

I appreciate the opportunity to review the revised version of the manuscript titled "Master-slave Game Optimal Scheduling for Multi-agent Integrated Energy System Based on Uncertainty and Demand Response," submitted to SUSTAINABILITY. Overall, the authors have responded well to my comments, making significant changes marked in red and providing additional explanations in relevant subsections, effectively addressing my concerns.

While the authors have acknowledged the importance of addressing the dynamic nature of Integrated Energy Systems (IES) in their future work, they could have provided more specific plans for tackling this aspect. Nonetheless, it is commendable that they have included this consideration in their future research agenda.

It is encouraging to see that the authors have recognized the significance of carbon emissions reduction and have promptly added references to support this aspect of their research. The clear overview of the carbon trading mechanism, emphasizing its role in controlling carbon emissions, is appreciated.

The authors’ utilization of Monte Carlo random sampling and clustering methods for scenario-based analysis is effective in allowing stakeholders to assess the energy system's performance under various conditions and optimize system operation accordingly. By evaluating revenue in each scenario, stakeholders can make informed decisions that maximize economic benefits while addressing environmental objectives, including carbon emissions' reduction.

While the authors have mentioned the use of YALMIP and CPLEX for quadratic programming, they have not explicitly stated which specific genetic algorithm they used, as noted in lines 431-432. Clarifying this aspect would enhance the understanding of the employed methodologies.

The information on the software tools used for mathematical modeling and optimization, including the Yalmip LMI parser and CPLEX, is insightful. It would be beneficial to briefly compare the efficiency of YALMIP and CPLEX with other optimization libraries available in various programming languages, such as CVXPY, MOSEK, or Gurobi, for a more comprehensive understanding.

The parameters used in the authors’ simulation, including electric, thermal, and cold loads, as well as outdoor and indoor temperatures, based on data from Northeast China, add realism to the simulation and enhance its applicability to real-world scenarios. Additionally, incorporating users’ preference coefficients for energy types reflects real-world demand variations and consumer behavior, contributing to the model’s accuracy.

In terms of amendments:

·        In Figure 3, at line 447, there appears to be a typographical error with the term “begining.” This should be corrected for clarity.

·        Consider comparing the efficiency of YALMIP and CPLEX with other optimization libraries available in various programming languages, such as CVXPY, MOSEK, or Gurobi, for a more comprehensive assessment.

·        Given the extensive mathematical modeling presented, it might be beneficial to focus more on the Monte Carlo iterations’ generation of different revenue scenarios, which aligns more closely with sustainability objectives.

I have included additional references that may provide valuable insights into multi-objective optimization and related topics.

Thank you once again for the opportunity to review this manuscript.

Best regards, 

References:

Basu, S., Mondal, A., Basu, A. (2019). A Cooperative Co-evolutionary Approach for Multi-objective Optimization. In: Bhattacharyya, S., Mukherjee, A., Bhaumik, H., Das, S., Yoshida, K. (eds) Recent Trends in Signal and Image Processing. Advances in Intelligent Systems and Computing, vol 727. Springer, Singapore. https://doi.org/10.1007/978-981-10-8863-6_7

Chen, J., Zhang, Q., Li, G. (2021). MOEA/D for Multiple Multi-objective Optimization. In: Ishibuchi, H., et al. Evolutionary Multi-Criterion Optimization. EMO 2021. Lecture Notes in Computer Science(), vol 12654. Springer, Cham. https://doi.org/10.1007/978-3-030-72062-9_13

Fujisawa, K. (2014). High Performance Computing for Mathematical Optimization Problem. In: Nishii, R., et al. A Mathematical Approach to Research Problems of Science and Technology. Mathematics for Industry, vol 5. Springer, Tokyo. https://doi.org/10.1007/978-4-431-55060-0_30

Li, W., Yuan, CM. (2019). Elimination Theory in Differential and Difference Algebra. J Syst Sci Complex 32, 287–316. https://doi.org/10.1007/s11424-019-8367-x

Löfberg, J. (2008). Modeling and solving uncertain optimization problems in YALMIP. IFAC Proceedings Volumes, Volume 41, Issue 2, pp. 1337-1341, ISSN 1474-6670. ISBN 9783902661005. https://doi.org/10.3182/20080706-5-KR-1001.00229

Löfberg, J. (2004). YALMIP: a toolbox for modeling and optimization in MATLAB. 2004 IEEE International Conference on Robotics and Automation (IEEE Cat. No.04CH37508), Taipei, Taiwan, pp. 284-289, doi: 10.1109/CACSD.2004.1393890.

Miguel Antonio, L., Coello Coello, C.A., Morales, M.A.R., Brambila, S.G., González, J.F., and G. C. Tapia (2020). Coevolutionary Operations for Large Scale Multi-objective Optimization. 2020 IEEE Congress on Evolutionary Computation (CEC), Glasgow, UK, pp. 1-8, doi: 10.1109/CEC48606.2020.9185846.

Miguel Antonio, L., Molinet Berenguer, J.A., and C.A. Coello Coello (2018). Evolutionary many-objective optimization based on linear assignment problem transformations. Soft Computing, 22:5491–5512 https://doi.org/10.1007/s00500-018-3164-3

Miguel Antonio, L., Coello Coello, C.A. (2016). Decomposition-Based Approach for Solving Large Scale Multi-objective Problems. In: Handl, J., Hart, E., Lewis, P., López-Ibáñez, M., Ochoa, G., Paechter, B. (eds) Parallel Problem Solving from Nature – PPSN XIV. PPSN 2016. Lecture Notes in Computer Science(), vol 9921. Springer, Cham. https://doi.org/10.1007/978-3-319-45823-6_49

Parrilo, P. (2003). Semidefinite programming relaxations for semialgebraic problems. Math. Program., Ser. B 96, 293–320. https://doi.org/10.1007/s10107-003-0387-5

Ritt J. (1932). Differential Equations from the Algebraic Standpoint. New York: Amer Math Soc.

Ritt J. (1950). Differential Algebra. New York: Dover Publications.

Small, S.M., and B. Jeyasurya (2007). Multi-Objective Reactive Power Planning: A Pareto Optimization Approach. 2007 International Conference on Intelligent Systems Applications to Power Systems, Kaohsiung, Taiwan, pp. 1-6, doi: 10.1109/ISAP.2007.4441630.

Wang, H., Sun, C., Yu, H. et al. (2022). A decomposition-based many-objective evolutionary algorithm with optional performance indicators. Complex Intell. Syst. 8, 5157–5176. https://doi.org/10.1007/s40747-022-00747-0

Web source a).  Multi-objective problems in YALMIP. Updated: November 24, 2017. https://yalmip.github.io/multi-objective-problems

Web source b).  Robust optimization. Updated: September 17, 2016. https://yalmip.github.io/tutorial/robustoptimization/

Author Response

Responses to Reviewer

Dear reviewer:

The authors would like to express their gratitude for your timely and competent reviews. We have carefully read the comments and revised our paper according to the comments and suggestions.

Response to Comments

Point 1: In Figure 3, at line 447, there appears to be a typographical error with the term begining. This should be corrected for clarity.

Response 1: The comment is greatly appreciated. Thank you for your kind suggestions. This typographical error has been corrected. And we have further checked the entire text and corrected the phrasing and grammatical errors in the paper. 

Point 2: Consider comparing the efficiency of YALMIP and CPLEX with other optimization libraries available in various programming languages, such as CVXPY, MOSEK, or Gurobi, for a more comprehensive assessment.

Response 2: The comment is greatly appreciated. Thank you for your kind suggestions. We have further explained and discussed YALMIP, CPLEX and other optimization libraries, which are marked in red in Sub-section 3.6.

Point 3: Given the extensive mathematical modeling presented, it might be beneficial to focus more on the Monte Carlo iterations generation of different revenue scenarios, which aligns more closely with sustainability objectives.

Response 3: The comment is greatly appreciated. Thank you for your kind suggestions. We have further strengthened the explanation and analysis of different revenue scenarios generated by Monte Carlo iterations, which are marked in red in Sub-section 4.2.
